# Topology-Matching Normalizing Flows for Out-of-Distribution Detection in Robot Learning

**Jianxiang Feng**[*,1] **Jongseok Lee**[2,3]**, Simon Geisler**[1]**, Stephan Günnemann**[1]**, Rudolph Triebel**[2,3]
[1] Department of Informatics, Technical University of Munich (TUM)
[2]Institute of Robotics and Mechatronics, German Aerospace Center (DLR)
[3] Department of Informatics, Karlsruhe Institute of Technology (KIT)
`jianxiang.feng@tum.de, {jongseok.lee, rudolph.triebel}@dlr.de,`
`{geisler, guennemann}@in.tum.de`

**Abstract:** To facilitate reliable deployments of autonomous robots in the real world, Out-of-Distribution (OOD) detection capabilities are often required. A powerful approach for OOD detection is based on density estimation with Normalizing Flows (NFs). However, we find that prior work with NFs attempts to match the complex target distribution topologically with naïve base distributions leading to adverse implications. In this work, we circumvent this topological mismatch using an expressive class-conditional base distribution trained with an information-theoretic objective to match the required topology. The proposed method enjoys the merits of wide compatibility with existing learned models without any performance degradation and minimum computation overhead while enhancing OOD detection capabilities. We demonstrate superior results in density estimation and 2D object detection benchmarks in comparison with extensive baselines. Moreover, we showcase the applicability of the method with a real-robot deployment.

**Keywords:** Normalizing Flows, Out-of-Distribution, Robotic Introspection

## 1 Introduction

The reliable identification of Out-of-Distribution (OOD) data, which is not well represented in the training set, poses a pressing challenge on the path towards trustworthy open-world robotic systems such as self-driving cars [1], delivery drones [2] or healthcare robots [3]. For example, with widespread adoption in the perception pipeline, existing object detectors have been reported to over-confidently misclassify an OOD object into a known class, which might obfuscate the decision-making module and eventually cause catastrophic consequences in safety-critical scenarios [1, 4, 5].

Normalizing Flows (NFs) are a popular class of generative models [6, 7, 8, 9] that may be used for OOD detection. NFs represent complex probability distributions [10] with a learnable series of transformations from a simple base distribution to a complex target distribution. However, NFs' expressivity [11, 12, 13] and numerical stability [14, 15] is limited by a fundamental constraint: the supports of the base and target distribution should preserve *similar topological properties* (Definition 3.3.10 in Runde [16]). The topological properties subsume different geometrical characteristics of the target distribution, including its continuity, the number of connected components, or the number of modes. Increasing the capacity of the transformation may mitigate this constraint. Yet, this raises computation and memory demands [11, 17, 12]. An alternative to overcome the topological mismatch is to increase the flexibility of the base distribution, which is surprisingly under-explored in the OOD detection literature.

Therefore, we propose to equip NFs with efficient but flexible base distributions for OOD detection in robot learning. Concretely, we replace the frequently used uni-modal Gaussian base distribution

---

[*]: work done when working at DLR.
code: https://github.com/DLR-RM

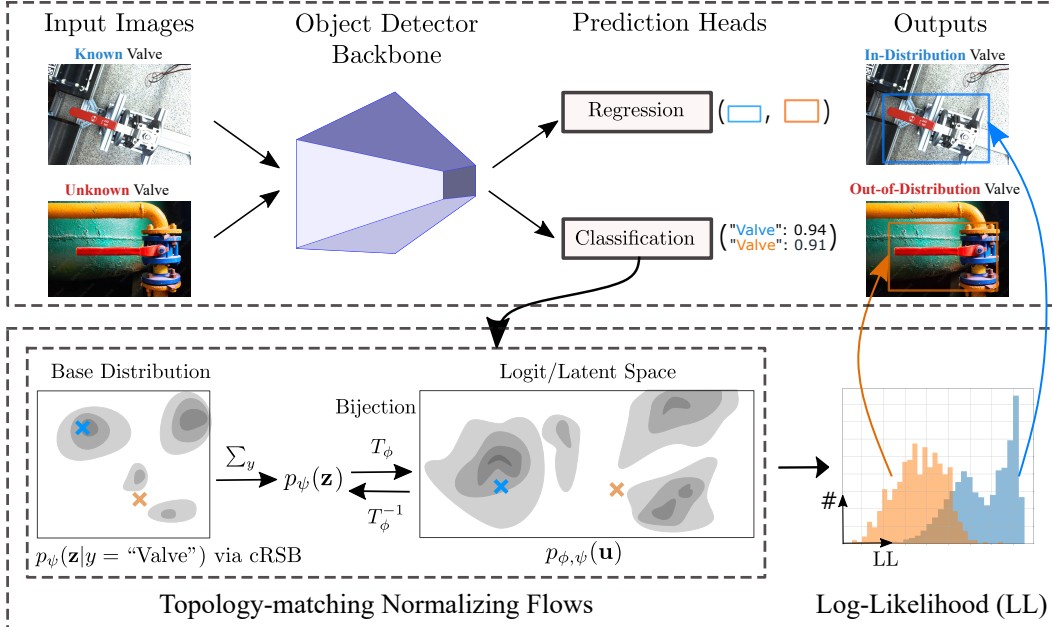

Figure 1: **The proposed architecture**. We overcome the topological mismatch problem in NFs to accurately model In-Distribution (ID) density. That is, the Conditional Resampled Base Distributions (cRSB) base distribution trained with Information Bottleneck (IB) $p_\psi(\mathbf{z}|\mathbf{y})$ can, e.g., adapt the numbers of modes to match target distribution with complex topology. Then we can identify OOD objects by low predicted log-likelihoods more reliably (best viewed in color).

with the cRSB, a class-conditional version of a learnable base distribution for mitigating the topological problem in NFs – Resampled Base Distributions (RSB) [13]. cRSB can learn the required topological properties, like adapting the number of modes, to match the unknown topological structure of the latent class-specific target distribution (Figure 1). Moreover, we adapt our cRSB with an adapted IB objective [18] to balance fusing class-conditional information with the marginalized density estimation capabilities in NFs. IB [19] is an information-theoretic objective to incorporate task-specific details e.g. class conditions, which are commonly ignored in pure generative modeling. This delivers a topology in the base distribution that is more accurately aligned to the one in the target distribution (see Figure 3).

Our OOD detection approach using topology-matching NFs is powerful and yet resource-efficient for open-set object detection. It is applicable to diverse object detectors (e.g., Faster-RCNN [20] and Yolov7 [21] used in this work) with minor changes and no loss of prediction performance. Moreover, our approach is sampling-free, i.e., only a single forward pass is required for efficient test-time inference while keeping the space memory tractable. As a result, our method is suitable for robotic applications that require a fast and robust perception module. We empirically show the state-of-the-art performance of the proposed idea using synthetic density estimation and 2D object detection tasks against extensive baselines. To further validate the applicability in robotics, we examine an object detector equipped with the proposed method on an exemplary inspection and maintenance aerial robot, showing the practical benefits of negligible memory and run-time overhead.

**Contributions.** Our main contribution is a NFs-based OOD detection method that overcomes the topological constraints while taking class-conditional information into account. We show that training with IB yields effective representation with superior OOD detection capabilities. We conduct a comprehensive empirical evaluation using both synthetic density estimation and public object detection datasets followed by a real-world robot deployment, which overall shows the effectiveness of the proposed approach.

## 2 Methodology

**Problem Formulation**    Given an image $\mathbf{x} \in \mathcal{X}$ and a trained object detector $F_\theta$ that localizes a set of objects with corresponding bounding box coordinates $\mathbf{b}_i \in \mathcal{R}^4$ as well as class label $y_i \in \mathcal{Y} = \{1, 2, ..., C\}$, the task is to distinguish if $(\mathbf{x}, \mathbf{b}_i, y_i)$ is ID, i.e., drawn from $\mathcal{P}_{id}$, or OOD, i.e., belongs to the unknown distribution $\mathcal{P}_{ood}$. For conciseness, from now on we omit the suffix $i$ and use $y$ to denote the class label without further notice. As discussed, a powerful OOD detection can be obtained via density estimation using NFs. This density estimator identifies OOD objects with low likelihoods after being trained *only* on data drawn from $\mathcal{P}_{id}$. Following relevant prior [22, 23], we use the semantically rich logit space (pre-softmax layer) for density estimation. To note that, our method can be readily applied to other (high-dimensional) latent feature spaces.

NFs are known to be universal distribution approximators [10]. That is, they can model a complex target distribution $p(\mathbf{u})$ on a space $\mathcal{R}^d$ by defining $\mathbf{u}$ as a transformation $T_\phi : \mathcal{R}^d \rightarrow \mathcal{R}^d$ from a well-defined base distribution $p_\psi(\mathbf{z})$, where $\phi$ and $\psi$ are model parameters, respectively:

$$\mathbf{u} = T_\phi(\mathbf{z}) \text{ where } \mathbf{z} \sim p_\psi(\mathbf{z}) \quad (1)$$

where $\mathbf{z} \in \mathcal{R}^d$ and $p_\psi$ is commonly chosen as a uni-modal Gaussian. By designing $T_\phi$

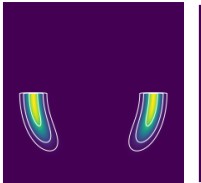 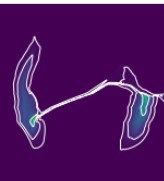 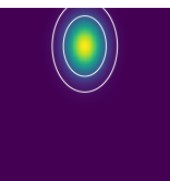

(a) $p(\mathbf{u}|y = 0)$    (b) $p_{\phi,\psi}(\mathbf{u}|y=0)$    (c) $p_\psi(\mathbf{z}|y = 0)$

Figure 2: Filament connect modes in the modeled class-conditional distribution (b) if using (trainable) uni-modal base (c) for the multi-modal target (a).

to be a *diffeomorphism*, that is, a bijection where both $T_\phi$ and $T_\phi^{-1}$ are differentiable, We can compute the likelihood of the input $\mathbf{u}$ *exactly* based on the change-of-variables formula [24]:

$$p_{\phi,\psi}(\mathbf{u}) = p_\psi(T_\phi^{-1}(\mathbf{u}))| \det(J_{T_\phi^{-1}}(\mathbf{u}))| \,, \quad (2)$$

where $J_{T_\phi^{-1}}(\mathbf{u}) \in \mathcal{R}^{d \times d}$ is the Jacobian of the inverse $T_\phi^{-1}$ with respect to $\mathbf{u}$. When the target distribution is unknown but samples thereof are available, we can estimate the parameter $(\phi, \psi)$ by minimizing the forward Kullaback-Leibler Divergence (KLD), which is equivalent to maximizing the expected Log-Likelihood (LL).

**Topological Mismatch**    However, since the base distribution $p_\psi(\mathbf{z})$ is usually a uni-modal Gaussian (e.g. Figure 2c) and $T_\phi$ is a diffeomorphism, problems arise for modeling data distribution with different topological properties. These include well-separated multi-modal distributions or distributions with disconnected components (e.g., Figure 2a). For example, one can see that this leads to density filaments between the modes in Figure 2b. Cornish et al. [11] have shown that flows require a bijection with *infinite bi-Lipshitz constant* when modeling a target distribution with disconnected support using a unimodal base distribution. Besides the diminishing modeling performance, this renders the bijection to be numerically "non-invertible", thus, causing optimization instability during training and unreliability of likelihood calculation [14].

### 2.1  Conditional Resampled Base Distributions

One possible partial mitigation is by enriching the expressiveness of the flows. For example, by (a) increasing the number of layers or parameters, (b) using more complex base distributions, or (c) employing multiple NFs, e.g., mixtures of NFs. It is important to note that especially (a) and (c) may escalate the computational cost and memory burden. Moreover, scaling the normalizing flow's expressivity, (a) or (c), often does not increase the stability of the optimization [15] or the likelihood calculation. For these reasons, we pursue (b) and attempt to compensate for the complexity of the transformation with the elasticity of the base distribution. In other words, we use a more flexible but efficient base distribution to trade off a costly but sufficiently expressive bijection of the normalizing flow. This way we aim to capture desirable topological properties of the target distribution [17]. Following the prior work [25], to model the fidelitous distribution of data with task-specific conditions, e.g. class labels, we use a class-conditional base distribution. This way we get similar benefits like combining multiple conditional flows (c), however, without having to burden the computational

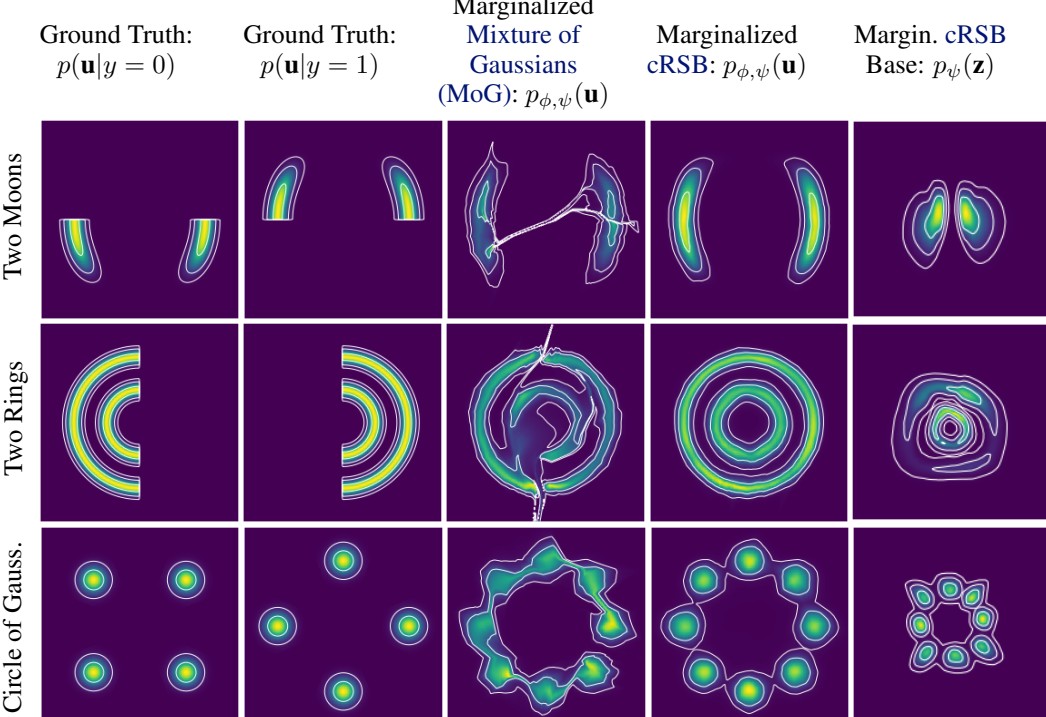

Figure 3: Visualization of density estimation using Real NVP with class conditional MoG, where each class is modeled by a uni-modal Gaussian, and cRSB as well as the class-marginalized density for the base distribution of cRSB.

cost on marginalization over classes. This is because, with (c), this operation requires repeated evaluation of the flows when each flow of the NFs mixture is class-conditional [26]. Even though a class-conditional distribution can specialize on a smaller fraction of the dataset containing similar instances, it will manifest in a multi-modal distribution.

Therefore, we propose to capture the complex topological properties in the target distribution with a more expressive base distribution instead of the uni-model Gaussian. To the end, we introduce cRSB by extending a powerful unconditional base distribution RSB [13] with class-conditional modeling. RSB deforms a uni-modal Gaussian in a learnable manner to obtain more complex distributions via Learned accept/reject sampling (LARS) [27]. LARS iteratively re-weighs samples drawn from a proposal distribution $\pi(\mathbf{z})$, e.g. a standard Gaussian, through a learned acceptance function $a_\psi : \mathcal{R}^d \to [0, 1]$. To reduce the computation cost in practice, this process is truncated by accepting the $T$-th samples if the previous $T - 1$ samples get rejected. To take into account class-conditional information, we conditionalize the learnable acceptance function $a_\psi(\mathbf{z}|y)$. As a result, we have the conditional base distribution:

$$p_\psi(\mathbf{z}|y) = (1 - \alpha_T)\frac{a_\psi(\mathbf{z}|y)\pi(\mathbf{z})}{Z_y} + \alpha_T\pi(\mathbf{z}), \tag{3}$$

where $a_\psi : \mathcal{R}^d \to [0, 1]^C$ and $\alpha_T = (1 - Z_y)^{T-1}$, where $Z_y \in \mathcal{R}$ is the normalization factor for $a_\psi(\mathbf{z}|y)\pi(\mathbf{z})$. This factor can be estimated via Monte Carlo Sampling.

In Figure 3, we contrast the density estimation capabilities of NFs with the common MoG [8, 25] base distribution and our cRSB on three tasks with class-conditional structure using an appropriate learning objective (see next section). We find that our cRSB learns appropriate topology-matching base distributions (right outer column) and as a result, the respective NFs do not have adverse effects like filaments between the modes.

## 2.2 Training with Information Bottleneck

Unfortunately, directly training NFs with a conditional base distribution can lead to underperformance as observed in experiments (see Table 2 and appendix) and reported by Fetaya et al. [25].

We attribute this to the lack of explicit control for the balance between generative and discriminative modeling in the likelihood-based training objective of NFs. To alleviate this, we train the normalizing flow with a class-conditional base distribution using the IB objective [19]. To abuse the notations, we denote random variables by capital letters such as $U, Z, Y$, and their realizations by lowercase letters such as $\mathbf{u}, \mathbf{z}, y$. The IB minimizes the Mutual Information (MI) $I(U, Z)$ between $U$ and $Z$, while simultaneously maximizing the MI $I(Z, Y)$ between $Z$ and $Y$. Intuitively, the IB trades off between the objectives of modeling the class conditional information $p(\mathbf{u}|y)$ with the marginalized density $p(\mathbf{u})$, thus allowing to leverage the class-conditional structure to facilitate more effective density estimation for data characterized with semantic classes.

However, the IB is not directly applicable to latent class-conditional distributions in NFs since the bijection $T_\phi$ is lossless by design. Thus, for trading off the class-conditional information with density estimation capabilities, we adapt the approach proposed by Ardizzone et al. [18] for our cRSB. Specifically, we inject a small amount of noise $\epsilon$ into the input $U$ and hence $Z_\epsilon = T_\phi^{-1}(U + \epsilon)$. Further we define an asymptotically exact version of MI, namely the Mutual Cross-Information (CI) (more details in appendix):

$$\mathcal{L}_{\text{IBNF}} = CI(U, Z_\epsilon) - \beta CI(Z_\epsilon, Y) \tag{4}$$

$$CI(U, Z_\epsilon) = \mathbb{E}_{p(\mathbf{u}), p(\epsilon)} \left[ -\log \sum_{y\prime} p_\psi(\mathbf{z}_\epsilon|y\prime) - \log|\det(J_{T_\phi^{-1}}(\mathbf{u} + \epsilon))| \right], \tag{5}$$

$$CI(Z_\epsilon, Y) = \mathbb{E}_{p(y)} \left[ \log \frac{p_\psi(\mathbf{z}_\epsilon|y)p(y)}{\sum_{y\prime} p_\psi(\mathbf{z}_\epsilon|y\prime)p(y\prime)} \right], \tag{6}$$

where $\mathbf{z}_\epsilon = T_\phi^{-1}(\mathbf{u} + \epsilon)$, $p(\epsilon) = \mathcal{N}(0, \sigma^2 \mathcal{I}_d)$ is a zero-meaned Gaussian with variance $\sigma^2$, and $\beta$ trades off class information and generative density estimation. With flexible conditional base distributions defined in Eq. 3, we can train the *topology-matching* NFs with IB by substituting cRSB into the conditional base probability $p_\psi(\mathbf{z}|y)$ in Eq. 5 and 6. More noteworthy, we observed that the IB is able to regularize the acceptance rate learning for cRSB to better assimilate the topological structure of the target distribution, leading to an overall improved performance on accurately approximating the complex target distribution (see Figure 4).

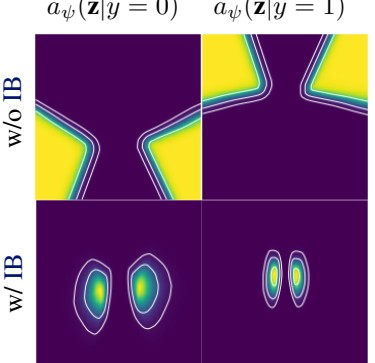

Figure 4: cRSB acceptance rate $a_\psi(\mathbf{z})$ w/o and w/ IB training for Two Moons.

### 2.3 Detecting OOD Objects

During test time, we detect the OOD data based on the predicted Log-Likelihood (LL). To note that, only one forward pass is required to evaluate the acceptance function in cRSB. Practically, we use Monte Carlo sampling to estimate the normalization factor $Z$ offline so that no additional computation required for this during inference. We *marginalize* the density over classes for the base distribution defined in Eq. 3 and compute the final LL given the logits $\mathbf{u}'$ from the test image:

$$\text{LL}_{test}(\mathbf{u}') = \log \sum_{y\prime} (p_\psi(T_\phi^{-1}(\mathbf{u}')|y\prime)) + \log|\det(J_{T_\phi^{-1}}(\mathbf{u}'))|. \tag{7}$$

We then expect LL for ID objects to be higher than OOD ones.

## 3 Related Work

**Normalizing Flows** NFs [28] are a popular class of deep generative models. NFs have shown applicability in a variety of areas such as image generation [29, 30], uncertainty estimation [31, 32, 33] and OOD detection [6, 34, 35]. For NFs, one trend has been designing expressive flow-based architectures. Notable examples are affine coupling flows [29, 30], auto-regressive flows [36, 37], invertible ResNet blocks [38] and ODEs-based maps [39]. The major focus of these works is on reducing computing requirements for Jacobian computations while ensuring that each mapping is

invertible. Another research direction, currently emerging, is on addressing the topological mismatch [28, 10] of NFs. Targeting this problem, some existing works attempt to increase the learning capacity of the transformation via mixture models [26], latent variable models [11, 40] or injecting carefully specified randomness [41, 12]. These methods may be limited in their applicability to robotics because they either increase memory consumption by expanding the width of transformations or approximate the exact likelihood. Recently, these constraints have been addressed by improving the expressivity of the base distribution [13, 17]. In this paper, we build upon this class of methods since they only add slight computation overheads and thus are well suited for applications in robotics.

**Normalizing Flows for OOD Detection** NFs have been widely adapted for OOD detection due to its superior density estimation [42]. For example, though with some counter-intuitive observations on raw data space [34], NFs have demonstrated encouraging OOD detection results with additional refinements for raw data [43, 44, 45] or directly based on task-relevant feature embeddings [6, 7, 46, 47]. In this work, we directly apply NFs on the feature space. To note that, another principle direction is to estimate the error bound for this task [48]. Recently hybrid models [49, 7, 50] have shown remarkable performance gain on OOD detection by modeling the joint distribution of both data and its class labels. Such works suggest that class labels can provide useful information. However, directly performing class conditional modeling with NFs for OOD detection results in performance degradation. Tishby et al. [19], Ardizzone et al. [18] mitigate such performance degradation by utilizing IB for training NFs. This explicitly controls the trade-off between generative and discriminative modeling [9]. However, these works on OOD detection utilize NFs without much concern for the fundamental topological problem as the first citizen. Therefore, complementary to these approaches, we examine the problem of topological mismatch of NFs for OOD detection.

**OOD Detection in Object Detectors** OOD detection research has focused on image classification [42], which may be limited in relevance to robotic vision. In robotics, we may often need both categorization and localization of objects of interest. Therefore, we focus on object detection in open-set conditions here. In this domain, uncertainty estimation [51] has been considered propitious for OOD detection but suffered from computation burdens on runtime [52, 53, 54, 55] or memory costs [56]. To address this, instead of directly applying uncertainty estimation techniques for object detection [54, 2], another popular approach is to explicitly formulate the problem as OOD detection tasks [23, 57, 8, 58, 59]. Amongst them, NFs has been utilized as an expressive density estimator [8, 58]. However, despite the encouraging results, these approaches have not examined the problem of topological mismatch in NFs. As this might prevent additional performance improvements, this work examines the topology-matching NFs for OOD detection in object detectors.

## 4 Experiments

We next demonstrate the efficacy of our method. First, we evaluate on synthetic density estimation for distributions with distinct topological properties. We then evaluate the OOD detection performance on two object-detection data-sets adapted from their public counterparts [60, 61] for open-set (OS) experiments: Pascal-VOC-OS and MS-COCO-OS based on Glow [30] and a pre-trained Faster-RCNN [20] provided by Miller et al. [23] for a fair comparison. To showcase the practicality, we deploy the one-stage object detector Yolov7 [21] equipped with the proposed method on a real aerial manipulation robot along with the run-time and memory analysis. We empirically found that, to parameterize the acceptance function in LARS, a simple multi-layer perceptron (MLP) (2x128 for density estimation and 3x128 for object detection) is sufficient. We select the hyper-parameters (e.g., $T, \epsilon, \sigma, \beta$) based on the validation set. More details can be found in the supplementary materials.

**Datasets and Metrics** For density estimation, there are three synthetic datasets: two moons, two rings, and a circle of Gaussians. We employ the KLD between the target and the model distributions to measure the performance. For OOD detection, since existing object detection datasets are not ready for fair evaluation [4], we strictly follow the experimental protocol in [23]. For real robot deployment, we generate $2k$ synthetic images of two objects (a valve and a crawler robot) rendered

based on their CAD models and additionally labeled $2k$ real images. $1k$ synthetic images are used for training and another $1k$ for testing with all real images. We use the Area Under Receiver Operation Curve (AUROC) and the True Positive Rate(TPR) at different False Positive Rate (FPR) ($5\%$, $10\%$, $20\%$) as metrics for this task, as they represent the performance of the potential operating points for safety-critical applications, which requires the FPR to be sufficiently low.

## 4.1 Density Estimation

We compare the density estimation performance in Table 1 and provide qualitative results in Figure 3. We find that the cRSB base distribution consistently outperforms the class-conditional Mixture of Gaussians (MoG). The performance improvement by cRSB can be generalized across two different NFs architectures, i.e. Real NVP and NSFs.

Table 1: Performance on density estimation for different flow architectures w.r.t. KLD, i.e., $D_{\mathrm{KL}}(p(\mathbf{u}, y) || p_{\phi,\psi}(\mathbf{u}, y))$. Better base distribution is highlighted in bold.

| Flow architecture | Real NVP | | NSFs | |
|---|---|---|---|---|
| Base distribution | MoG_IB | cRSB_IB | MoG_IB | cRSB_IB |
| Two Moons | 1.179 | **1.066** | 0.909 | **0.906** |
| Two Rings | 2.032 | **1.704** | 1.647 | **1.602** |
| Circle of Gaussians | 2.335 | **1.667** | 1.766 | **1.653** |

## 4.2 OOD Detection in Object Detection

We compare our method (cRSB_IB) with both flow-based and non-flow-based approaches. The latter consists of Mahalanobis Distance (MD) [62], Relative Mahalanobis Distance (RMD) [63], GMMDet [23], Softmax, Entropy and, their Deep Ensemble variants with five models [56]. Among flow-based approaches, we have six different base distributions, including unconditional ones (uni-modal Gaussian, MoG, RSB) and their conditional variants

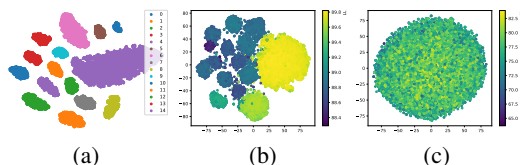

(a)      (b)      (c)

Figure 5: t-SNE visualization for (a) feature embeddings from the object detector (b) latents of the proposed learned base distribution cRSB and (c) the uni-modal Gaussian on the training set of Pascal-VOC-OS.

(MoG_CLS, cRSB_CLS) [25] and MoG trained with IB (MoG_IB) [8, 18]. From Table 2, we can observe that flows with uni-modal Gaussian are able to provide satisfactory performance, i.e., better than most of non flow-based baselines, while flows with more expressive base distributions such as MoG and RSB can bring more benefits on Pascal-VOC-OS than MS-COCO-OS. When trained with IB, the more flexible conditional base distribution (cRSB_IB) can mostly have greater performance gains (on both Pascal VOC and COCO) than its strong competitor (MoG_IB) (only on COCO) in comparison with their counterparts without IB (MoG_CLS). These results demonstrate the effectiveness of cRSB with IB for OOD detection in complicated 2D object detection tasks. We further provide the visualization from data before and after the flow transformation with different base distributions in Figure 5, evidencing the ability of matching complex topology of the target data distribution with cRSB.

## 4.3 Real Robot Deployment

Next, we validate the applicability in an application of robotic inspection and maintenance, where it is crucial to avoid false positives of OOD objects that appear routinely in outdoor environments. In this experiment, we train Yolov7 with only synthetic images of two objects (a valve and a crawler robot) and deploy on the robot around only real objects. The task is to identify the falsely detected real objects as OOD since they are from a distribution different to the synthetic ones. Besides, the performance drop when compared with Table 2 is potentially attributed to the "closer" OOD data because the synthetic images are rendered in a highly photorealistic manner. However, our method still outperform other baseline approaches in Figure 6c, where ours can notably achieve higher TPR around the low FPR region, which are commonly used as operating points for the robot. Computational efficiency is another important requirement. We compare the runtime and space memory consumption against a vanilla Yolov7 using the NVIDIA's embedded GPU called Jetson Orin in Figure 6. The results indicate that the computational overhead of having an OOD detector is

Table 2: OOD detection performance on Pascal-VOC-OS and MS-COCO-OS datasets for different methods based on the Faster-RCNN from 3 random runs. The highest values are marked in **bold** and the second highest in *italics*.

| | Pascal-VOC-OS | | | | MS-COCO-OS | | | |
| --- | --- | --- | --- | --- | --- | --- | --- | --- |
| | AUROC | TPR at | | | AUROC | TPR at | | |
| | | 5%FPR | 10%FPR | 20%FPR | | 5%FPR | 10%FPR | 20%FPR |
| Softmax | 0.901 | 60.1 | 72.8 | 83.1 | 0.882 | 61.3 | 70.6 | 78.1 |
| Entropy | 0.905 | 59.8 | 72.9 | 82.9 | 0.903 | 61.2 | 70.6 | 80.2 |
| MD [62] | 0.9 | 54.1 | 68.8 | 83.3 | 0.902 | 57.2 | 71.4 | 85.5 |
| RMD [63] | 0.838 | 15.2 | 28.4 | 77.4 | 0.531 | 1.7 | 2.6 | 7.1 |
| Ensemble Softmax [56] | 0.885 | 47.8 | 72.6 | 83.1 | 0.898 | 66.2 | 73.5 | 82.3 |
| Ensemble Entropy [56] | 0.887 | 47.8 | 72.5 | 83.1 | 0.906 | 66.2 | 73.5 | 82.3 |
| GMMDet [23] | 0.931 | 70.7 | *80.5* | *89.3* | 0.924 | 69.5 | 80.2 | 87.9 |
| Flows Gaussian | 0.915 ± 0.002 | 72.2 ± 0.75 | 77.8 ± 0.89 | 86.1 ± 0.67 | 0.924 ± 0.001 | 68.2 ± 0.73 | 81.2 ± 0.61 | 89.4 ± 0.04 |
| Flows MoG | 0.919 ± 0.002 | 69.0 ± 2.4 | 77.0 ± 2.5 | 86.5 ± 1.2 | 0.925 ± 0.001 | 68.3 ± 0.30 | 80.5 ± 0.50 | 89.6 ± 0.05 |
| Flows RSB [13] | 0.924 ± 0.003 | 72.8 ± 0.88 | 79.3 ± 1.0 | 87.1 ± 0.82 | 0.925 ± 0.001 | 68.6 ± 0.87 | 81.3 ± 0.31 | 89.5 ± 0.34 |
| Flows MoG_CLS [25] | 0.923 ± 0.001 | 69.2 ± 1.5 | 78.2 ± 1.3 | 88.5 ± 0.82 | *0.930* ± 0.001 | 68.5 ± 0.73 | *82.2* ± 0.31 | *89.7* ± 0.30 |
| Flows MoG_IB [8] | *0.934* ± 0.002 | 73.1 ± 1.3 | 79.6 ± 0.6 | 87.8 ± 0.2 | 0.924 ± 0.002 | *71.1* ± 0.9 | 79.6 ± 0.46 | 88.6 ± 0.63 |
| Flows cRSB_CLS | 0.919 ± 0.001 | 72.5 ± 0.37 | 78.8 ± 0.27 | 86.8 ± 0.42 | 0.924 ± 0.001 | 68.3 ± 0.14 | 81.1 ± 0.30 | 89.3 ± 0.18 |
| Flows cRSB_IB (ours) | **0.946** ± 0.003 | **78.5** ± 0.97 | **84.0** ± 0.83 | **90.8** ± 0.76 | **0.934** ± 0.002 | **73.3** ± 2.0 | **84.3** ± 0.40 | **91.3** ± 0.28 |

relatively small when compared to the vanilla Yolov7. Overall, these experiments validate our claim that our method features efficient runtime inference and cost-effective memory consumption.

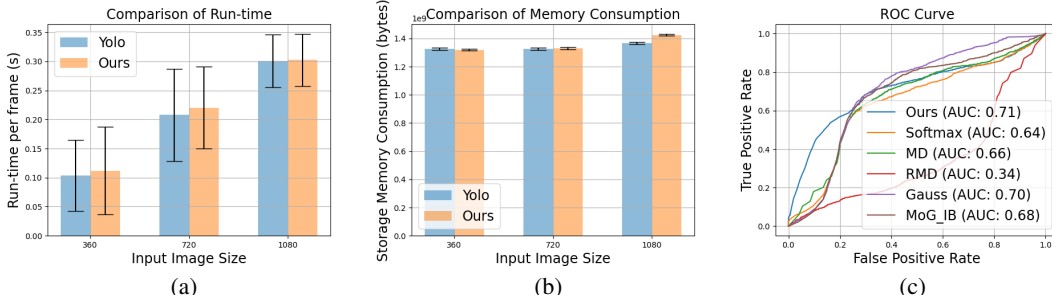

Figure 6: Results from experiments on a real robot. Run-time, memory consumption, and ROC curve are reported. Compared to the vanilla Yolov7 , the proposed method does not yield significant computational costs, while providing performance gains in OOD detection.

## 5 Limitations

The proposed method is envisioned to work on feature embeddings instead of raw data to counteract the NFs artifacts of assigning higher likelihoods to OOD data [10]. This leads to two limitations. First, it's can't directly applied to the tasks/models that could not provide useful feature embeddings extracted from the raw data. Second, its performance is restricted to the quality of features. As reported by previous work [23, 8], learning more compact and centralized features can often lead to increased performance for OOD detection while feature collapse can be harmful to OOD detection. Besides, there are two limitations during deployment. The first is the prolonged initialization time for calculating the normalization factor in LARS based on Monte Carlo sampling. This might not be friendly for applications that require instant response at the beginning. Moreover, the current version of the proposed method does not consider the sequential nature of observations at deployment.

## 6 Conclusion

To endow robots with introspective awareness against OOD data, we propose the NFs equipped with effective yet lightweight cRSB and train with IB objective. Such NFs are able to mitigate the fundamental topological mismatch problem, facilitating more effective OOD detection capabilities. We present empirical evidence that the proposed method achieves superior performance both quantitatively and qualitatively. To demonstrate the run-time efficiency and minimum memory overheads, we deployed on a real-robot system. Overall, we hope that the results of our work stemming from an enriched base distribution can push forward the direction of NFs-based OOD detection in robot learning.

**Acknowledgments**

We thank the anonymous reviewers for their thoughtful feedback. Jianxiang Feng and Simon Geisler are supported by the Munich School for Data Science (MUDS). Rudolph Triebel and Stephan Gunnemann are members of MUDS.

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

# Supplementary Materials for the Submission: Topology-Matching Normalizing Flows for Out-of-Distribution Detection in Robot Learning

**Jianxiang Feng**[*,1] **Jongseok Lee**[2,3]**, Simon Geisler**[1]**, Stephan Günnemann**[1]**, Rudolph Triebel**[2,3]
[1] Department of Informatics, Technical University of Munich (TUM)
[2]Institute of Robotics and Mechatronics, German Aerospace Center (DLR)
[3] Department of Informatics, Karlsruhe Institute of Technology (KIT)
jianxiang.feng@tum.de, {jongseok.lee, rudolph.triebel}@dlr.de,
{geisler, guennemann}@in.tum.de

## 1 Background

### 1.1 Resampled Base Distribution

Learned Accept/Reject Sampling (LARS) approximates a d-dimensional distribution $q(z)$ by reweighting a proposal distribution. The reweightings are learned through a parametrized acceptance function. The parameter is $\phi$, which determines the shape of the acceptance function. If a sample of $\pi$ is given, it is accepted with a certain probability, otherwise, it is rejected, and a new sample is drawn until one of the proposed samples is accepted. This means:

$$p_\infty(z) = \frac{\pi(z)a_\phi(z)}{Z} \quad \text{with} \quad Z := \int_\pi (z)a_\phi(z)dz. \tag{1}$$

One can also reduce the reject rates by a truncation parameter. The trick here is to accept the T-th sample, if the first T-1 samples are rejected. This is independent of the value provided by LARS. With this, the sampling distribution becomes, given $\alpha_T := (1 - Z)^{T-1}$ :

$$p_T(z) = (1 - \alpha_T)\frac{a_\phi(z)\pi(z)}{Z} + \alpha_T\pi(z) \quad \text{where} \quad Z \approx \frac{1}{S}\sum_{s=1}^{S} a_\phi(z_s). \tag{2}$$

We note that reducing rejection rates are desirable in order to reduce the computational overhead. Z is often approximated due to intractability. As parameters of the acceptance function cause changes in Z, $z_s$ needs to be recomputed in every training iteration.

The resample base distributions (RSB) relies on LARS. LARS can be used as a proposal to avoid topological mismatches of the flows. Then, density could become the aforementioned sampling distribution. The log-likelihood using such distributions can be derived as:

$$\log p(x) = \log \pi(z) + \log\left(\alpha_T + (1 - \alpha_T\frac{a_\phi(z)}{Z})\right) - \log|\det J_{F_\theta}(z)|. \tag{3}$$

$F_\theta$ is the flow transformation. The intuition is that by learning the parameters of the base distribution while keeping it computationally feasible with LARS, the base distribution is modified so that the topology mismatch problem can be addressed [1].

7th Conference on Robot Learning (CoRL 2023), Atlanta, USA.

## 1.2 Information Bottleneck for Normalizing Flows

The Information Bottleneck (IB) for Normalizing Flows (NFs) with class conditional base distribution, and its derivation, closely follows Ardizzone et al. [2]. They originally derived this quantity for the class-conditional Mixture of Gaussians (MoG).

Generally, the IB [3] is defined as

$$\mathcal{L}_{\text{IB}} = I(U, Z) - \beta I(Z, Y) \tag{4}$$

with Mutual Information (MI) $I$ and trade-off parameter $\beta$. To derive the learning objective presented in the main part two key steps are required: (1) we substitute MI $I$ with the so-called *mutual cross-information $CI$*, and (2) we inject Gaussian noise in the otherwise lossless transformation between $U$ and $Z$.

The *mutual cross-information $CI$* [2] is defined as

$$CI(A, B) = \mathbb{E}_{a,b \sim p(A,B)} \left[ \log \frac{q(a,b)}{q(a)q(b)} \right] \tag{5}$$

where $q$ represents the approximative distribution of the true density $p$. We then replace both occurrences of $I$ in Equation (4) with $\hat{I}$. Moreover, for simplicity, we assume that $q(Y) = p(Y)$, and in our experiments, we model $p(Y)$ uniformly.

The bijection $T_\phi$ is loss-less by design and, thus, the joint distributions $p(U, Z)$ and $p(U, Z)$ are not valid Radon-Nikodym densities rendering $I$ and $\hat{I}$ undefined. To circumvent the issue and as indicated above (2), we artificially introduce noise $\epsilon \sim \mathcal{N}(0, \sigma^2 \mathbf{I})$ and model $Z_\epsilon = T_\phi^{-1}(U + \epsilon)$ instead. Then, dropping all terms that are independent of the model or vanish for $\sigma \to 0$, we obtain (see Ardizzone et al. [2] for full details)

$$CI(U, Z_\epsilon) = \mathbb{E}_{p(\mathbf{u}),p(\epsilon)} \left[ -\log \sum_{y\prime} (p_\psi(\mathbf{z}|y\prime)) - \log |\det(J_{T_\phi^{-1}}(\mathbf{u} + \epsilon))| \right], \tag{6}$$

Similarly, $CI(Y, Z_\epsilon)$ resolves to

$$CI(Y, Z_\epsilon) = \underbrace{\mathbb{E}_{p(y)} \left[ -\log p(y) \right]}_{= \text{const.}} + \mathbb{E}_{p(Z,Y),p(\epsilon)} \left[ \log \frac{p_\psi(T_\phi^{-1}(\mathbf{u} + \epsilon)|y)p(y)}{\sum_{y\prime}(p_\psi(T_\phi^{-1}(\mathbf{u} + \epsilon)|y\prime)p(y\prime))} \right]$$

$$= \mathbb{E}_{p(Z,Y),p(\epsilon)} \left[ \log \frac{p_\psi(\mathbf{z}|y)p(y)}{\sum_{y\prime}(p_\psi(\mathbf{z}|y\prime)p(y\prime))} \right] + \text{const.} \tag{7}$$

with $\mathbf{z} = T_\phi^{-1}(\mathbf{u} + \epsilon)$. The overall objective combines Equation (6) and Equation (7) to

$$\mathcal{L}_{\text{IBNF}} = CI(U, Z) - \beta CI(Z, Y) \tag{8}$$

In summary, with $CI(U, Z)$ we minimize the marginalized density $p(\mathbf{u})$ subject to the noise term $\epsilon$. On the other hand, $\mathcal{L}_{\text{IBNF}}$ optimizes for predictive performance, again subject to noise $\epsilon$.

## 2 Implementation Details

### 2.1 Synthetic Density Estimation

We follow a resource constraint setup to allow for conclusions about practical mobile applications. Specifically, we use a Real NVP [4] with four layers for the cRSB and five layers for the MoG with randomly initialized trainable mean and variance. The difference in layers accounts for the extra compute that is required for the cRSB. For the NSF we use two layers each since the cRSB overhead is negligible here. In the MoG's we model a class-conditional Gaussian $\mathcal{N}(\boldsymbol{\mu}_c, \boldsymbol{\sigma}_c \mathbf{I})$ with diagonal variance matrix.

Table 1: Implementation Details for density estimation.

| hyper-params | cRSB | MoG |
|---|---|---|
| IB loss | $\beta(0.1); \sigma(0.01)$ | $\beta(0.1); \sigma(0.01)$ |
| learning rate | $1e-3$ | $1e-3$ |
| training epoch | $10,000$ | $25,000$ |
| optimizer | Adamax | Adamax |
| batch_size | 1024 | 1024 |

## 2.2 Out-of-Distribution (OOD) detection for 2D Object Detection

For object detection, we implement our method with the Glow [5] architecture based on the *norm-flows* open-sourced package [6]. Besides, we also ablate on this with different ones such as Neural Spline Flows (NSF) [7]. On object detection task, we train our flows based on logits generated by the pre-trained Faster-RCNN [8] object detectors provided by Miller et al. [9] for fair comparison [1]. We summarize the training details for the results presented in the main paper in Table 2 and Table 3. Regarding the trainability for the base distribution in MoG, we found minor difference between setting it trainable and untrainable and the major impacting factor for performance improvement and training stability is the distance between means for initialization. Therefore we seek to tune this hyper-parameter and leave the base distribution untrainable.

We compare with the following baseline approaches:

- Non Flow-based:
    - Softmax
    - Entropy
    - Mahalanobis Distance (MD)[2] [10]
    - Relative Mahalanobis Distance (RMD) [11]
    - Softmax Ensemble (5) [12]
    - Entropy Ensemble (5) [12]
    - GMMDet [9].

- Flow-based:
    - Flows Gaussian
    - Flows MoG
    - Flows Resampled Base Distributions (RSB) [1]
    - Flows MoG_CLS [13]
    - Flows cRSB_CLS [13]
    - Flows MoG_IB [14, 2].

Table 2: Implementation Details for Glow on Pascal-VOC-OS.

| hyper-params | cRSB_IB | GMM_IB | cRSB_CLS | GMM_CLS | RSB | GMM | Gaussian |
|---|---|---|---|---|---|---|---|
| base | Dropout(0.1); $T(100)$; $\epsilon(0.05); a(\cdot) : 3 \times 128$; | distance-scale for means initialization:10 | Dropout(0.1); $T(100)$; $\epsilon(0.05); a(\cdot) : 3 \times 128$; | distance-scale for means initialization:10 | Dropout(0.1); $T(100)$; $\epsilon(0.01); a(\cdot) : 3 \times 128$; | distance-scale for means initialization:10 | — |
| flow arch | $16 \times [4 \times 64]$ | $16 \times [4 \times 64]$ | $16 \times [4 \times 64]$ | $16 \times [4 \times 64]$ | $16 \times [4 \times 64]$ | $16 \times [4 \times 64]$ | $16 \times [4 \times 64]$ |
| IB loss | $\beta(30.0); \sigma(0.5)$ | $\beta(50.0); \sigma(0.5)$ | — | — | — | — | — |
| learning rate | $1e-4$ | $1e-4$ | $1e-4$ | $1e-4$ | $1e-4$ | $1e-4$ | $1e-4$ |
| training epoch | 400 | 400 | 200 | 400 | 200 | 400 | 400 |
| optimizer | Adam | Adam | Adam | Adam | Adam | Adam | Adam |
| batch_size | 1024 | 1024 | 1024 | 1024 | 1024 | 1024 | 1024 |

---

[1]https://github.com/dimitymiller/openset_detection
[2]https://github.com/stanislavfort/exploring_the_limits_of_OOD_detection

Table 3: Implementation Details for Glow on MS-COCO-OS.

| hyper-params | cRSB_IB | GMM_IB | cRSB_CLS | GMM_CLS | RSB | GMM | Gaussian |
|---|---|---|---|---|---|---|---|
| base | Dropout(0.1); $T(100)$; $\epsilon(0.01)$; $a(\cdot): 3 \times 128$; | distance-scale for means initialization:10 | Dropout(0.1); $T(100)$; $\epsilon(0.05)$; $a(\cdot): 3 \times 128$; | distance-scale for means initialization:10 | Dropout(0.1); $T(100)$; $\epsilon(0.01)$; $a(\cdot): 3 \times 128$; | distance-scale for means initialization:10 | — |
| flow arch | $8 \times [4 \times 128]$ | $8 \times [4 \times 128]$ | $8 \times [4 \times 128]$ | $8 \times [4 \times 128]$ | $8 \times [4 \times 128]$ | $8 \times [4 \times 128]$ | $8 \times [4 \times 128]$ |
| IB loss | $\beta(30.0); \sigma(0.1)$ | $\beta(50.0); \sigma(0.5)$ | — | — | — | — | — |
| learning rate | $1e-4$ | $1e-4$ | $1e-4$ | $1e-4$ | $1e-4$ | $1e-4$ | $1e-4$ |
| training epoch | 200 | 200 | 200 | 200 | 200 | 200 | 200 |
| optimizer | Adamax | Adamax | Adamax | Adamax | Adamax | Adamax | Adamax |
| batch_size | 512 | 1024 | 1024 | 1024 | 1024 | 1024 | 1024 |

## 2.3 Real Robot Deployment

We used the open-sourced implementation [3] for training and testing the object detector yolov7 [15] based on datasets described in Section 3.3. Specifically, we trained the architecture of yolov7-e6e with a learning rate $1e-2$, weight decay $5e-4$, batch size 2, image size $720 \times 720$ and SGD optimizer for 50 epochs. The detector was then deployed on the embedded computing module NVIDIA Jetson Orin on our aerial manipulation robot.

An aerial manipulation system is composed of a mobile platform, capable of moving in the 3D world. Carrying a robotic manipulator, such systems extend the mobility of robotic manipulators. Several applications are envisioned. Amongst them, in this paper, we ground our method in applications of robotic inspection and maintenance. The perception system has to understand its surroundings semantically, and here, deep learning-based methods are the current golden standards. Unfortunately, learning-based methods often assume that the test samples are generated from the same distribution as the training data. This assumption is routinely violated in the real world, and out-of-distribution detectors aim to identify such failure cases of learning-based methods.

For the implementation details, our in-house developed robot consists of one stereo camera, one monocular camera, one RBG-D camera, and a LiDAR sensor. In this work, since the semantics of the scenes may rely on vision as its main modalities, we utilize one monocular camera to detect the objects of interest, which are an industrial valve, and an inspection robotic crawler for oil and gas pipes in refineries. For computing, the robot is equipped with two NVIDIA Jetson Orin. In the experiments adapted, the real images were captured in a mock-up facility, and tested with the NVIDIA Jetson ORIN on the robot. The experimental data were collected with 30W mode with JETPACK 5.1.1. Auvidea carrier board is used.

## 3 Datasets

In this section, we provide more details on the datasets used in the experiments.

### 3.1 2-D Density Estimation

In each training epoch, we sample a new batch according to the subsequent unnormalized log densities following Stimper et al. [1]. For **Two Moons**, we use

$$-\frac{(\|\mathbf{u}\| - 1)^2}{0.08} - \frac{(|u_0| - 2)^2}{0.18} + \log\left(1 + \mathrm{e}^{-\frac{4u_0}{0.09}}\right) \tag{9}$$

and define the class assignments in $y \in \{0, 1\}$ via $p(y|\mathbf{u}) = u_1 > 0$. In words, the class is 1 for positive $y$-values and 0 otherwise. The **Two Rings** distribution is defined as

$$\log\left[\sum_{i=1}^{2}\left(\frac{32}{\pi}\exp\left\{-32(\|\mathbf{u}\| - i - 1)^2\right\}\right)\right] \tag{10}$$

---

[3] https://github.com/WongKinYiu/yolov7

Table 4: Implementation Details for Glow on Yolov7 on the Robot.

| hyper-params | cRSB_IB | GMM_IB | Gaussian |
|---|---|---|---|
| base | Dropout$(0.1)$; $T(100)$; $\epsilon(0.05)$; $a(\cdot):3\times128$; | distance-scale for means initialization:0.1 | — |
| flow arch | $2\times[4\times64]$ | $2\times[4\times64]$ | $2\times[4\times64]$ |
| IB loss | $\beta(10.0)$; $\sigma(0.01)$ | $\beta(1.0)$; $\sigma(0.01)$ | — |
| learning rate | $1e-5$ | $1e-4$ | $1e-4$ |
| training epoch | 200 | 200 | 200 |
| optimizer | Adamax | Adamax | Adamax |
| batch_size | 128 | 128 | 128 |

where $p(y|\mathbf{u}) = u_0 > 0$. Here we split classes along the y-axis. Last, the **Circle of Gaussians** is given by

$$\log\left[\sum_{i=1}^{8}\left(\frac{9}{2\pi(2-\sqrt{2})}\exp\left\{-\frac{9\left(\left(u_0-2\sin\left(\frac{2\pi}{8}i\right)\right)^2+\left(u_1-2\cos\left(\frac{2\pi}{8}i\right)\right)^2\right)}{4-2\sqrt{2}}\right\}\right)\right] \quad (11)$$

where we assign each Gaussian plot in an alternating scheme to the classes $y \in \{0,1\}$. Concretely, we split the classes according to $\sin(4\cdot(\text{atan2}(u_1,u_0)+\frac{\pi}{8})) > 0$. For this, we use the $\text{atan2}(u_1,u_0)$ function that is available in many programming environments. We follow the IEEE convention for value combinations like $u_0 = u_1 = 0$.

### 3.2 OOD detection for 2D Object Detection

Following the experimental protocol used in [9], we first construct the open-set object detection data set as they did for Pascal-VOC [16] and MS-COCO [17], dubbed as Pascal-VOC-OS and MS-COCO-OS. In this setting, the first 15 classes and 50 classes are selected as known classes from the Pascal-VOC and MS-COCO datasets respectively, while the remaining classes are set as unknown classes. Therefore, the training sets are then filtered to *keep only known classes*. The original test sets are used for testing as they have both known and unknown classes. The idea behind is to exclude the effects of the unknown objects in the backgrounds for object detector training, since the object detector is inherently trained to ignore the background objects seen during training. We summarize the detailed information regarding the training, validation and test set of the adapted object detection datasets in Section 3.2. It contains the sizes of correct and false predictions from the detector, i.e. True Positives (TP) and False Positives (FP). The FP here are from the OOD data which should not be detected by the detector.

For convenience in training NFs, we further extract the features (logits in our case) from all the detections predicted from the object detector. We then filter out the detections with low quality and keep the meaningful detections by setting thresholds for the Intersection-Over-Union (0.5) and confidence score (0.2). The purpose behind is to simplify the problem and focus on only meaningful False Positive (FP) predictions predicted by the detector.

In this task, we want to identify the OOD objects that are falsely detected and classified as known objects by the detector. To evaluate this ability, we use the Area Under Receiver Operation Curve (AUROC) as a metric with varying thresholds. To note that we set the true positives as the correct predictions and false positives as the incorrect predictions.

Table 5: Information of Pascal-VOC-OS and MS-COCO-OS

|  | Pascal-VOC-OS | MS-COCO-OS |
|---|---|---|
| ID train set | first 15 classes from Pascal-Voc2007&2012 train | first 50 classes from MS-COCO2017 train |
| ID val set | first 15 classes from Pascal-Voc2007&2012 val | first 50 classes from MS-COCO2017 val |
| Test set | 20 classes from Pascal-Voc2007 test | 80 classes from MS-COCO2017 test |
| #TP in training set | 18318 | 251202 |
| #TP in val set | 7601 | 55593 |
| #ID FP in val set | 348 | 2287 |
| #TP in test set | 8288 | 16148 |
| #ID FP in test set | 213 | 784 |
| #OOD FP in test set | 660 | 1068 |

Figure 1: Exemplar images from training and test set used in the real robot experiment.

## 3.3 OOD detection on Real Robot

The synthetic images used for training in this part were generated by BlenderProc [18]. The test set contains real images collected in our lab. Some examplar images are shown in Figure 1. We summarize the sizes of the used training and test data in Section 3.3.

Table 6: Training and test data for real robot experiments

| #images in Sim train set | 907 |
|---|---|
| #images in Sim test set | 855 |
| #images in Real test set | 2078 |
| #ID True Positives in test set | 1326 |
| #OOD False Positives in test set | 1898 |

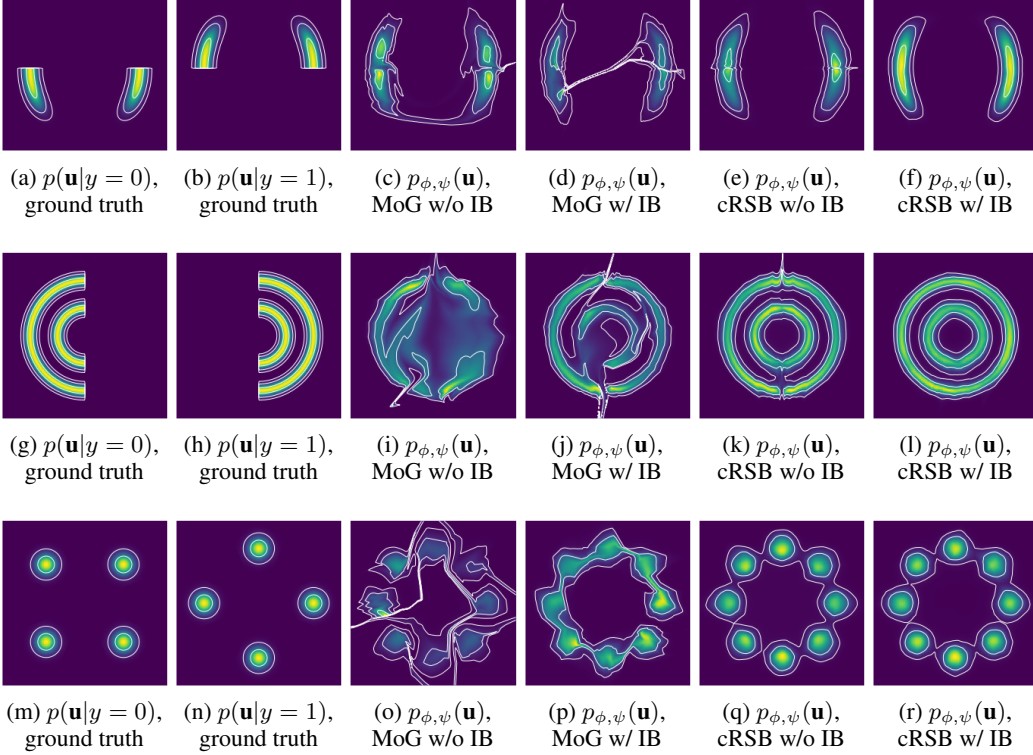

(a) $p(\mathbf{u}|y=0)$, ground truth  (b) $p(\mathbf{u}|y=1)$, ground truth  (c) $p_{\phi,\psi}(\mathbf{u})$, MoG w/o IB  (d) $p_{\phi,\psi}(\mathbf{u})$, MoG w/ IB  (e) $p_{\phi,\psi}(\mathbf{u})$, cRSB w/o IB  (f) $p_{\phi,\psi}(\mathbf{u})$, cRSB w/ IB

(g) $p(\mathbf{u}|y=0)$, ground truth  (h) $p(\mathbf{u}|y=1)$, ground truth  (i) $p_{\phi,\psi}(\mathbf{u})$, MoG w/o IB  (j) $p_{\phi,\psi}(\mathbf{u})$, MoG w/ IB  (k) $p_{\phi,\psi}(\mathbf{u})$, cRSB w/o IB  (l) $p_{\phi,\psi}(\mathbf{u})$, cRSB w/ IB

(m) $p(\mathbf{u}|y=0)$, ground truth  (n) $p(\mathbf{u}|y=1)$, ground truth  (o) $p_{\phi,\psi}(\mathbf{u})$, MoG w/o IB  (p) $p_{\phi,\psi}(\mathbf{u})$, MoG w/ IB  (q) $p_{\phi,\psi}(\mathbf{u})$, cRSB w/o IB  (r) $p_{\phi,\psi}(\mathbf{u})$, cRSB w/ IB

Figure 2: Comparison of generative modeling capabilities $p_{\phi,\psi}(\mathbf{u})$ including training w/o and w/ IB.

## 4 Density Estimation

In Figure 2, we provide a comparison of the (marginal) density estimation capabilities for the three density estimation datasets of NFs with MoG as well as Conditional Resampled Base Distributions (cRSB) base distributions, trained w/o as well as w/ IB. In Figure 3-6, we illustrate the marginal and class-conditional feature as well as base distributions.

The main conclusions are (1) training with IB objective results in better modeling capabilities of the marginal distribution $p_{\phi,\psi}(\mathbf{u})$; (2) training w/o IB objective not only results in more filaments between the modes but also causes spiky tendrils at areas where the class conditional distributions are touching; (3) results with our cRSB base distributions are essentially filament free; (4) training with IB allows for slight overlaps in the class conditional densities $p_{\phi,\psi}(\mathbf{u}|y)$ as well as $p_{\phi,\psi}(\mathbf{z}|y)$. In other words, here we improve the modeling capabilities of the marginal distribution $p_{\phi,\psi}(\mathbf{u})$ at the cost of imperfect modeling of the class-conditional structure.

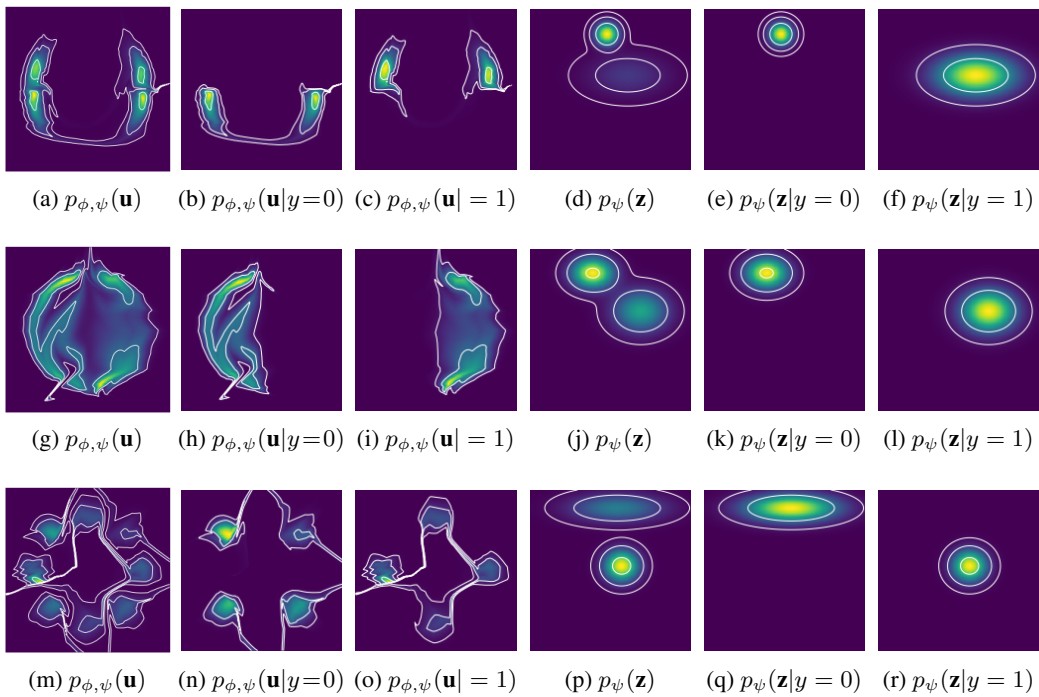

(a) $p_{\phi,\psi}(\mathbf{u})$    (b) $p_{\phi,\psi}(\mathbf{u}|y{=}0)$    (c) $p_{\phi,\psi}(\mathbf{u}|{=}1)$    (d) $p_{\psi}(\mathbf{z})$    (e) $p_{\psi}(\mathbf{z}|y=0)$    (f) $p_{\psi}(\mathbf{z}|y=1)$

(g) $p_{\phi,\psi}(\mathbf{u})$    (h) $p_{\phi,\psi}(\mathbf{u}|y{=}0)$    (i) $p_{\phi,\psi}(\mathbf{u}|{=}1)$    (j) $p_{\psi}(\mathbf{z})$    (k) $p_{\psi}(\mathbf{z}|y=0)$    (l) $p_{\psi}(\mathbf{z}|y=1)$

(m) $p_{\phi,\psi}(\mathbf{u})$    (n) $p_{\phi,\psi}(\mathbf{u}|y{=}0)$    (o) $p_{\phi,\psi}(\mathbf{u}|{=}1)$    (p) $p_{\psi}(\mathbf{z})$    (q) $p_{\psi}(\mathbf{z}|y=0)$    (r) $p_{\psi}(\mathbf{z}|y=1)$

Figure 3: Marginalized and class-conditional feature as well as base distributions for MoG trained w/o IB.

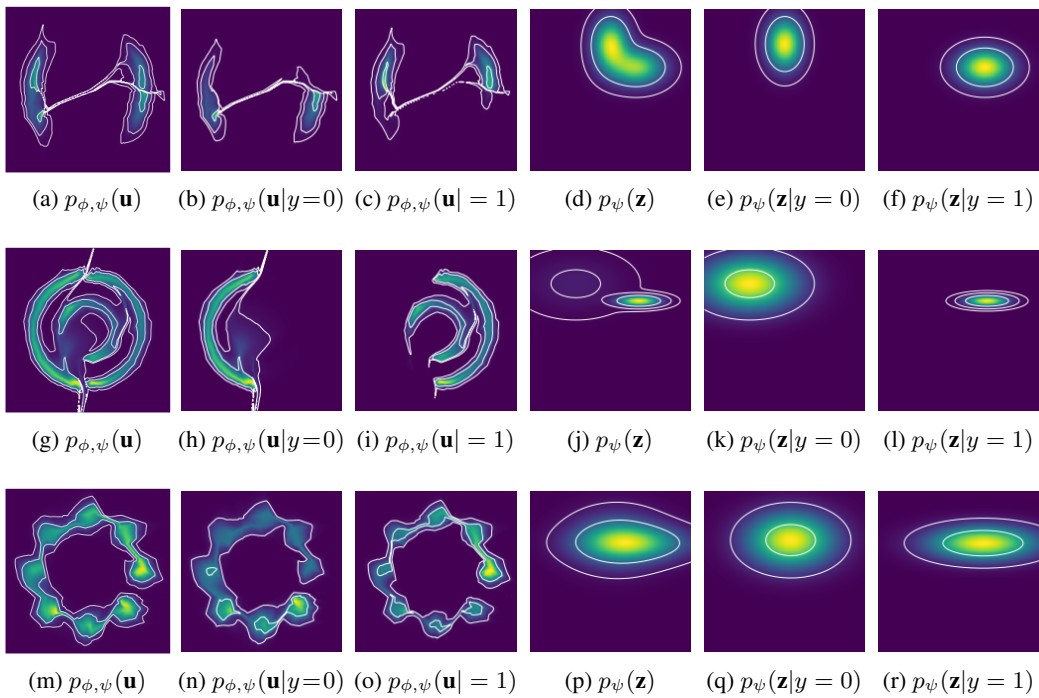

(a) $p_{\phi,\psi}(\mathbf{u})$    (b) $p_{\phi,\psi}(\mathbf{u}|y{=}0)$    (c) $p_{\phi,\psi}(\mathbf{u}|{=}1)$    (d) $p_{\psi}(\mathbf{z})$    (e) $p_{\psi}(\mathbf{z}|y=0)$    (f) $p_{\psi}(\mathbf{z}|y=1)$

(g) $p_{\phi,\psi}(\mathbf{u})$    (h) $p_{\phi,\psi}(\mathbf{u}|y{=}0)$    (i) $p_{\phi,\psi}(\mathbf{u}|{=}1)$    (j) $p_{\psi}(\mathbf{z})$    (k) $p_{\psi}(\mathbf{z}|y=0)$    (l) $p_{\psi}(\mathbf{z}|y=1)$

(m) $p_{\phi,\psi}(\mathbf{u})$    (n) $p_{\phi,\psi}(\mathbf{u}|y{=}0)$    (o) $p_{\phi,\psi}(\mathbf{u}|{=}1)$    (p) $p_{\psi}(\mathbf{z})$    (q) $p_{\psi}(\mathbf{z}|y=0)$    (r) $p_{\psi}(\mathbf{z}|y=1)$

Figure 4: Marginalized and class-conditional feature as well as base distributions for MoG trained w/ IB.

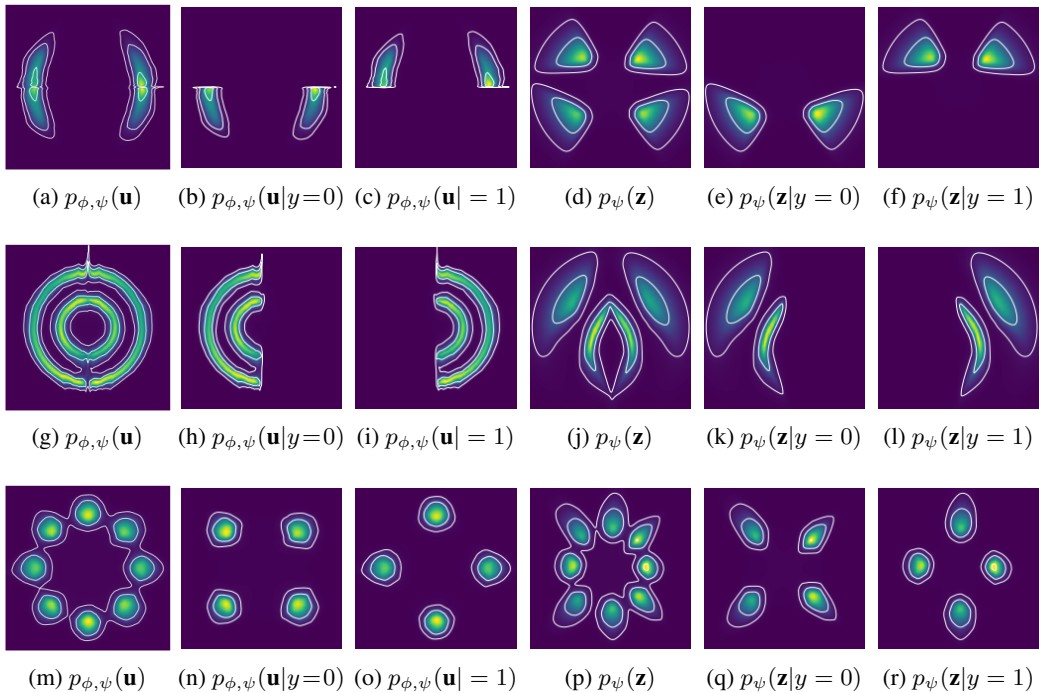

(a) $p_{\phi,\psi}(\mathbf{u})$    (b) $p_{\phi,\psi}(\mathbf{u}|y{=}0)$    (c) $p_{\phi,\psi}(\mathbf{u}|{=}1)$    (d) $p_{\psi}(\mathbf{z})$    (e) $p_{\psi}(\mathbf{z}|y=0)$    (f) $p_{\psi}(\mathbf{z}|y=1)$

(g) $p_{\phi,\psi}(\mathbf{u})$    (h) $p_{\phi,\psi}(\mathbf{u}|y{=}0)$    (i) $p_{\phi,\psi}(\mathbf{u}|{=}1)$    (j) $p_{\psi}(\mathbf{z})$    (k) $p_{\psi}(\mathbf{z}|y=0)$    (l) $p_{\psi}(\mathbf{z}|y=1)$

(m) $p_{\phi,\psi}(\mathbf{u})$    (n) $p_{\phi,\psi}(\mathbf{u}|y{=}0)$    (o) $p_{\phi,\psi}(\mathbf{u}|{=}1)$    (p) $p_{\psi}(\mathbf{z})$    (q) $p_{\psi}(\mathbf{z}|y=0)$    (r) $p_{\psi}(\mathbf{z}|y=1)$

Figure 5: Marginalized and class-conditional feature as well as base distributions for cRSB trained w/o IB.

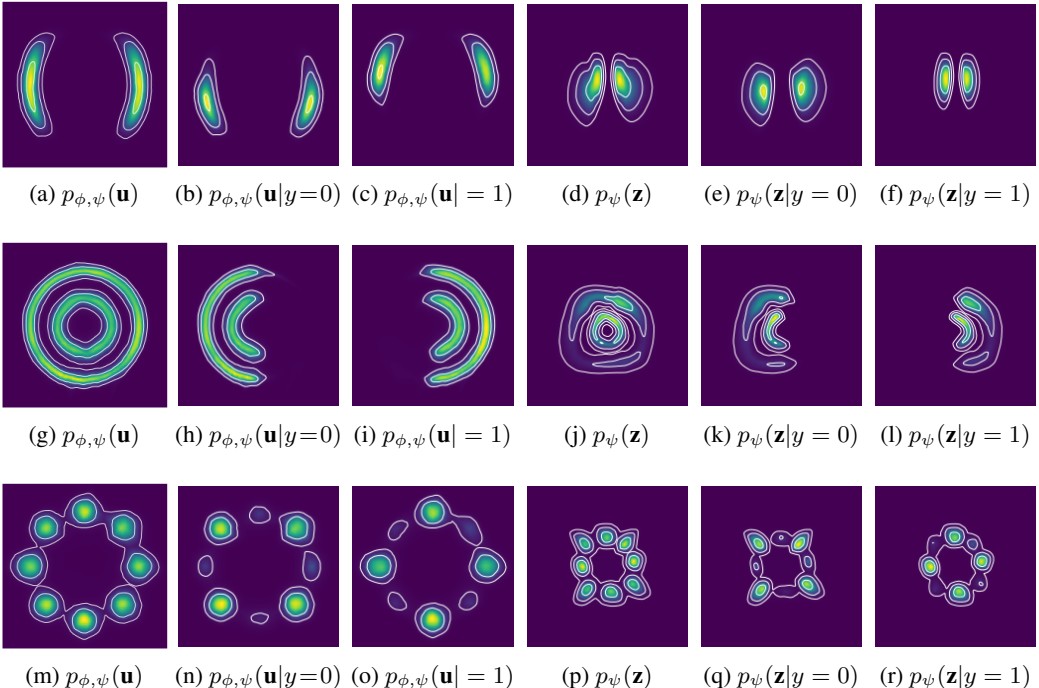

(a) $p_{\phi,\psi}(\mathbf{u})$    (b) $p_{\phi,\psi}(\mathbf{u}|y{=}0)$    (c) $p_{\phi,\psi}(\mathbf{u}|{=}1)$    (d) $p_{\psi}(\mathbf{z})$    (e) $p_{\psi}(\mathbf{z}|y=0)$    (f) $p_{\psi}(\mathbf{z}|y=1)$

(g) $p_{\phi,\psi}(\mathbf{u})$    (h) $p_{\phi,\psi}(\mathbf{u}|y{=}0)$    (i) $p_{\phi,\psi}(\mathbf{u}|{=}1)$    (j) $p_{\psi}(\mathbf{z})$    (k) $p_{\psi}(\mathbf{z}|y=0)$    (l) $p_{\psi}(\mathbf{z}|y=1)$

(m) $p_{\phi,\psi}(\mathbf{u})$    (n) $p_{\phi,\psi}(\mathbf{u}|y{=}0)$    (o) $p_{\phi,\psi}(\mathbf{u}|{=}1)$    (p) $p_{\psi}(\mathbf{z})$    (q) $p_{\psi}(\mathbf{z}|y=0)$    (r) $p_{\psi}(\mathbf{z}|y=1)$

Figure 6: Marginalized and class-conditional feature as well as base distributions for cRSB trained w/ IB.

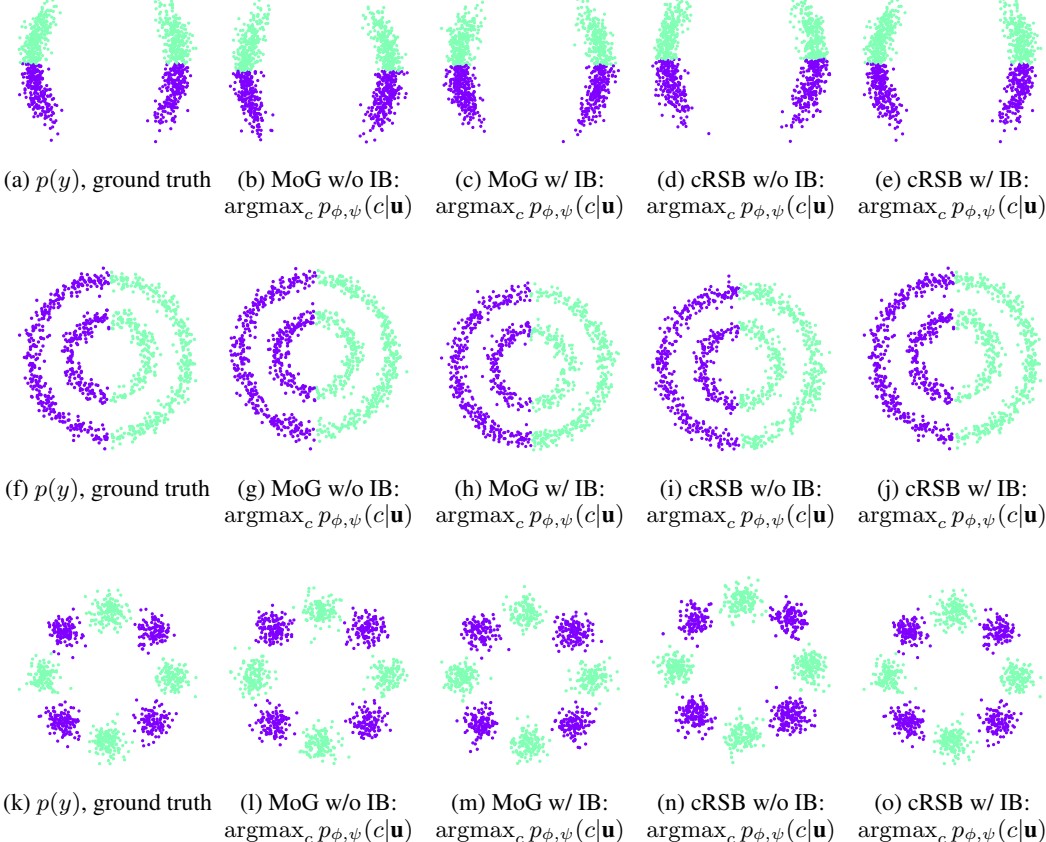

Figure 7: Comparison of predictive capabilities $\mathrm{argmax}_c\, p_{\phi,\psi}(c|\mathbf{u})$ including training w/o and w/ IB for a sample of size 1,024 that was drawn from the target distribution.

## 5 Generative Classification

In Figure 7, we plot the ground truth assignments for a sample drawn from the respective data distribution along with the most likely class assignment $\mathrm{argmax}_c\, p_{\phi,\psi}(c|\mathbf{u})$ over varying base distribution and training objectives. Despite the perceptually different results in the density estimation (Section 4), all models classify the input data remarkably well. This highlights the importance of the IB objective since it allows to govern the tradeoff between the quality of density estimation and predictive capabilities.

## 6 Ablation Study

### 6.1 Density estimation

In Figure 8, we study the influence of the tradeoff parameter $\beta$ on the density estimation quality. Generally, we observe that for small $\beta$ the class-conditional density is not modeled sufficiently, while too large values for $\beta$ result in a degradation of the quality of the marginal density. An exception is the Two Rings dataset, where a too strong emphasis on the class-conditional density degrades marginal and class-conditional density (i.e., too high $\beta$). We also experimented with different values for the noise defined by $\sigma$; however, we do not report detailed results here due to the low impact on the density estimation quality.

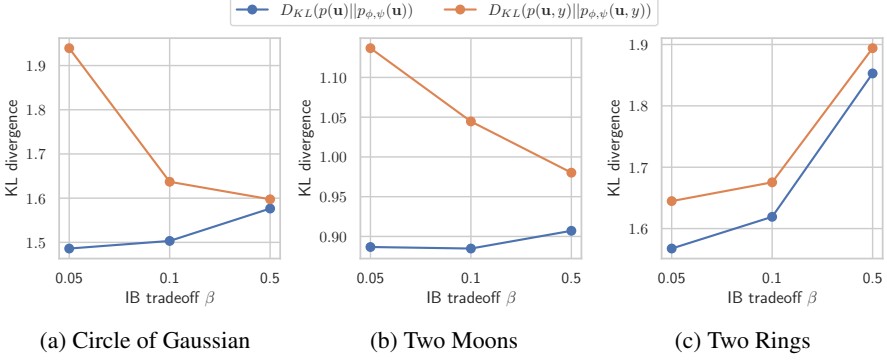

(a) Circle of Gaussian       (b) Two Moons       (c) Two Rings

Figure 8: Influence of hyperparemter $\beta$ on the marginalized density quality $D_{KL}(p(\mathbf{u})||p_{\phi,\psi}(\mathbf{u}))$ and class-conditional density quality $D_{KL}(p(\mathbf{u},y)||p_{\phi,\psi}(\mathbf{u},y))$.

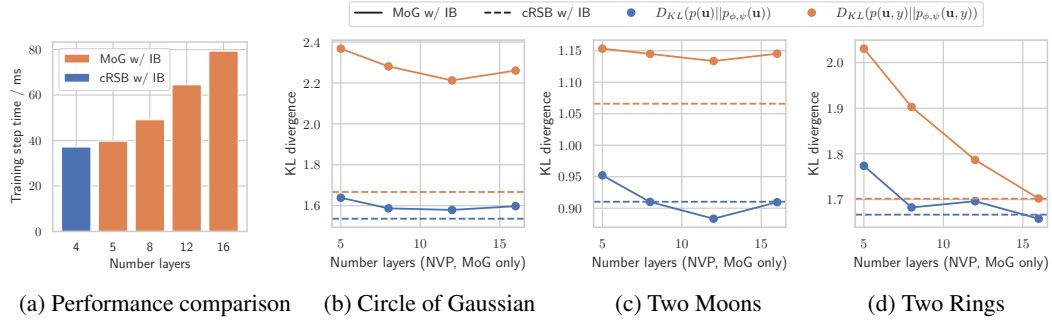

(a) Performance comparison    (b) Circle of Gaussian    (c) Two Moons    (d) Two Rings

Figure 9: Performance tradeoff between using the expressive base distribution cRSB vs. simple MoG. For cRSB we use 4 Real NVP layers. In the other density estimation experiments, we use 5 Real NVP layers with MoG base to account for the additional compute for the cRSB base. In (b), (c), and (d), we fix the number of layers to 4 if using the cRSB base distribution.

## 6.2   Computation Trade-off: Memory Costs and Training Runtime

**Training Runtime on Density Estimation:** In Figure 9, we study how easily a more expressive flow can compensate for the distribution mismatch. In the other density estimation experiments, we used 4 Real NVP layers with cRSB base and 5 Real NVP layers with MoG base. The different number of flow layers was chosen to account for the additional compute for the cRSB base. While more layers substantially impact runtime (see Figure 9a for a comparison of training runtime), more layers do not necessarily compensate for the less expressive MoG base distribution. Especially, class-conditional distribution modeling seems to rely heavily on the expressivity of cRSB. Moreover, for many layers, the training shows signs of increased instability.

**Memory Costs on OOD detection:** We present the ablation study on memory trade-off with Glow on MS-COCO-OS dataset in Table 7. The acceptance function is a MLP of 3x128. With this experiment, we observe that with a mixture of Gaussian, adding more layers escalate the memory costs ( 4 times). On the other hand, using a more expressive base distribution such as MoG and cRSB is able to yield similar performance but with less memory consumption ( 20%) than adding more layers. The savings would become more significant when the size of the NFs model is larger and the number of layers is higher.

## 6.3   $\beta$ and $\sigma$ in Information Bottleneck Loss

We present the ablation study on $\beta$ and $\sigma$ in Information Bottleneck Loss in Table 8. We used the Pascal-Voc-OS data set for this experiment. The truncation iteration T is set to 100 and running

Table 7: Computation Trade-off: Memory Costs on MS-COCO-OS

| Base distribution | Gaussian | MoG_IB | cRSB_IB |
|---|---|---|---|
| 2 | 0.918( 0.55Mb) | 0.916( 0.57Mb) | 0.921( 0.7Mb) |
| 8 | 0.924( 2.2Mb) | 0.924( 2.3Mb) | 0.934( 2.4Mb) |

average decay $\epsilon$ to 0.05. To note that, during loss computation, the trade-off parameter $\beta$ is converted to the weights for both terms with the following equations:

$$beta\_nll = 1./(1 + \beta) \tag{12}$$

and

$$beta\_cls = 1. * \beta/(1 + \beta). \tag{13}$$

This means that the larger $\beta$ is, the weights are larger for the class-discriminative term and smaller for the generative negative log-likelihood term. And $\epsilon$ controls the magnitude of noise injected to the input to induce information loss for a proper definition of mutual-cross-information. In Table 8, we can observe that first along the vertical dimension of the noisy variance, adding certain level of noise (e.g. 0.5) can improve the performance. But too much noise (e.g 5.0) corrupts the input and degrades the performance probably due to excessive information loss. On the other hand, $\beta$ controls the contribution of the class-conditional term, which includes the task-specific semantic class information. We can see by increasing this value, the performance boosts and gradually plateaus, implying the success of incorporating class-conditional information for more effective OOD detection.

Table 8: Ablation study on $\beta$ and $\sigma$ in Information Bottleneck Loss on Pascal VOC-OS for Glow.

| | $\beta$ | | | | |
|---|---|---|---|---|---|
| $\sigma$ | 1 | 5 | 10 | 30 | 50 |
| 0.1 | 0.930 | 0.938 | 0.945 | 0.944 | 0.944 |
| 0.5 | 0.937 | 0.946 | 0.948 | 0.951 | 0.940 |
| 1.0 | 0.929 | 0.949 | 0.944 | 0.940 | 0.937 |
| 5.0 | 0.702 | 0.096 | 0.717 | 0.118 | 0.134 |

### 6.4 Truncation Iteration $T$ and Running Average Decay $\epsilon$ in LARS

We present the ablation study on the truncation iteration T and running average decay $\epsilon$ in LARS in Table 9. We used the Pascal-Voc-OS data set for this experiment. To note the parameters in IB loss: $\beta$ is set to 30 and running average decay $\sigma$ to 0.5. In LARS, the truncation iteration $T$ controls the number of samples we need before we accept one and $\epsilon$ determines the running average decay for the normalization term computation during training. In Table 9, the changes of $T$ and $\epsilon$ lead to minor fluctuates in the final performance. We hypothesize that the minor difference from $\epsilon$ might be attributed to the re-computation of the normalization factor $Z$ during inference. The results in the table also show that there are only small impacts on the final performance for different values of $T$ and $\epsilon$ potentially because the acceptance function parameterized by a MLP plays a dominant role in the base distribution likelihood computation.

Table 9: Ablation study on T and $\epsilon$ on Pascal VOC-OS for Glow.

| | $\epsilon$ | | | | |
|---|---|---|---|---|---|
| $T$ | 0.001 | 0.01 | 0.05 | 0.1 | 0.5 |
| 50 | 0.941 | 0.941 | 0.950 | 0.953 | 0.945 |
| 100 | 0.944 | 0.948 | 0.951 | 0.953 | 0.947 |
| 250 | 0.945 | 0.949 | 0.949 | 0.946 | 0.946 |
| 500 | 0.929 | 0.945 | 0.948 | 0.949 | 0.950 |

## 6.5    TSNE-Visualization of Feature Embeddings and Flow Latents

In this section, we provide the t-SNE (T-distributed Stochastic Neighbor Embedding) visualization for inspecting how the proposed topology-matching NFs learn to match the target distribution. We used the Pascal-Voc-OS data set for this experiment. For this propose, we first visualize the feature embeddings extracted from the object detector in Figure 10a indexed by their class labels and the In-Distribution (ID) and OOD objects on Figure 10b. It can be observed that the feature embeddings possess a well class-separate structure and the OOD data is close to the ID data, which can be considered as near OOD data, a more challenging case [19] for OOD detection. Next, we visualize the transformed data in the latent space of the flows with different base distributions, namely cRSB and unimodal Gaussian in Figure 12 colored by class labels and Figure 11 colored by log-likelihoods. By comparison, it can be inspected that the latent structure of cRSB with IB can more faithfully resemble the target distribution in Figure 10. In contrast, the latent structure of unimodal Gaussian is fixed with all data points from different classes overlapping. The pieces of empirical evidence provide us the hint that a learnable base distribution cRSB with IB is able to match the complex topology of the target distribution more felicitous than an oversimplified one such as a unimodal Gaussian.

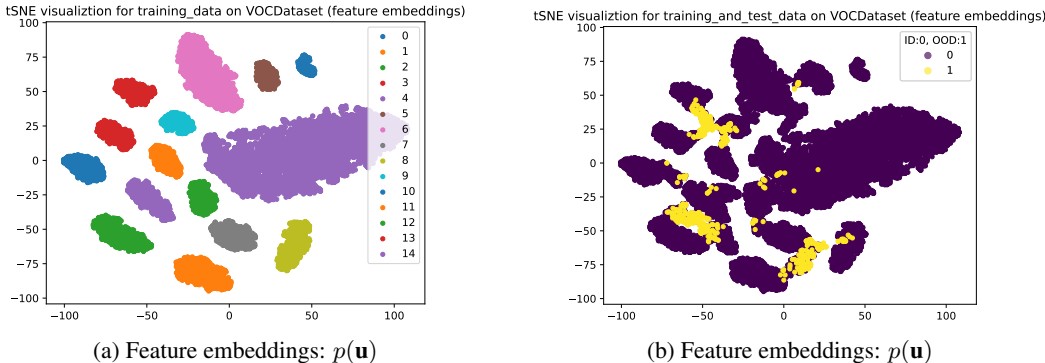

(a) Feature embeddings: $p(\mathbf{u})$         (b) Feature embeddings: $p(\mathbf{u})$

Figure 10: t-SNE visualization for (a) feature embeddings from the object detector on the training set (b) the feature embeddings of both ID and OOD data of Pascal-VOC-OS.

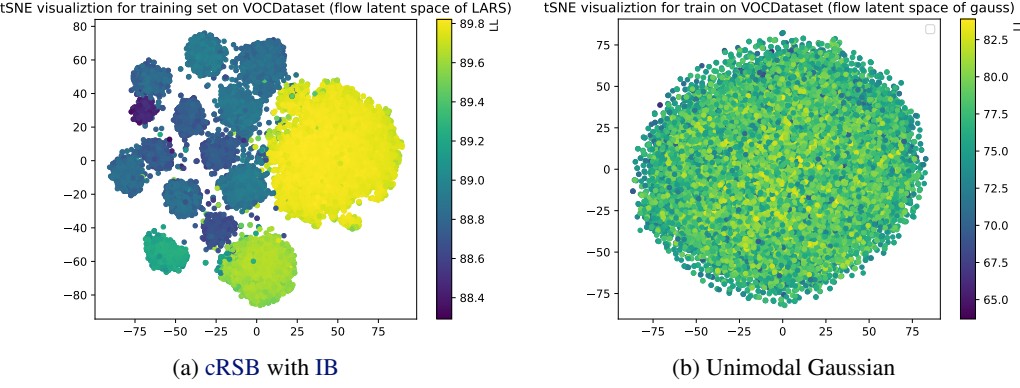

(a) cRSB with IB         (b) Unimodal Gaussian

Figure 11: t-SNE visualization for (a) latent features of the flow with cRSB (b) latent features of the flow with a uni-model Gaussian on the training set of Pascal-VOC-OS, valued by the log-likelihoods, respectively.

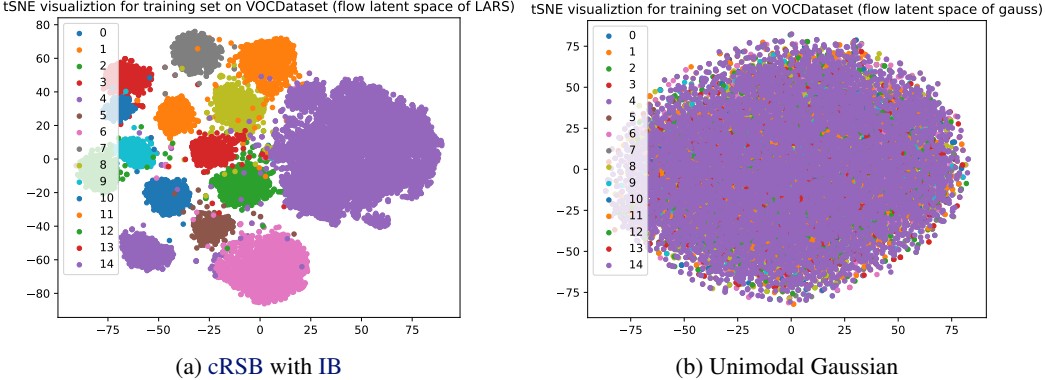

(a) cRSB with IB

(b) Unimodal Gaussian

Figure 12: t-SNE visualization for (a) latent features of the flow with cRSB (b) latent features of the flow with a uni-model Gaussian on the training set of Pascal-VOC-OS, indexed by the class memberships, respectively.

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
