# OpenReview forum: "Topology-Matching Normalizing Flows for Out-of-Distribution Detection in Robot Learning"
_robot-learning.org/CoRL/2023/Conference — CoRL 2023 Poster_

### Official Review · Reviewer_MswT · 2023-07-13

**Confidence:** 4
**Originality:** Fair
**Technical Quality:** Very Good
**Clarity Of Presentation:** Very Good
**Impact:** 3

**Recommendation:**

Weak Accept: I recommend accepting the paper, but will not argue for my recommendation if the majority of other reviewers have a different opinion.

**Review:**

Strengths
- This paper deals with an important issue for robot learning that is very relevant to the CoRL community
- The efficacy of the method is demonstrated convincingly by the results, with both benchmark results and a compelling real-robot deployment scenario
- I found the paper well written and the quality of exposition to be high.

Weaknesses
- Much of the cited literature, particularly [27, 35, 45] discusses how likelihood is often a poor choice both as training objective and as a means of performing OOD detection. This supports the author's use of the IB objective for training, but the authors still use the likelihood in order to perform OOD detection. I think the authors could engage more with this aspect of the literature. For instance, did the authors try other methods for performing OOD detection such as those in [45] or [46] for instance? Why is likelihood a good metric for OOD detection in their case?
- The main contributions of the paper are an extension of [13] to class-conditioned priors, and of [18] to the resampled base distribution. The technical contributions of the paper are fairly incremental.
- Limitations sections could be improved. Other than line 273, the limitations section is mostly justifying the method and arguing that the computational cost is negligible, rather than a frank assessment of the limitations. I would be interested to see the authors outline what tasks their current method is restricted to, or alternatively any limitations that might affect real-robot deployment. For instance, the current method does not consider the sequential nature of observations at deployment.

Misc questions
- What is the relative increase in computational cost due to the rejection sampling vs MoG flow with the same number of flow layers?


**Quality Of The Limitations Section:**

Additional details required

**Questions For Rebuttal:**

- Add more discussion / justification for the use of likelihood to perform OOD detection.
- Improve limitations section as mentioned above


**Robotics Focus:**

Sufficient demonstration on hardware

**Summary Of Paper:**

This paper presents a method for performing out-of-distribution (OOD) detection for trained object detection models. The authors use a Normalizing Flow to learn a class conditioned generative model of the object detector class classifier logits, and use the likelihood of this generative model to perform OOD detection. The main contributions of this paper are extending an existing work [13], which uses learned rejection sampling of the base distribution, to class conditioned base distributions. The authors also adapt the method from [18] in order to train this model with an information bottleneck (IB) objective to prevent class information being under-utilized. The authors show in experiments that their method outperforms several baselines on benchmark datasets on a variety of metrics, as well as demonstrate the method in the context of a robotic inspection task.

**Summary Of Recommendation:**

I believe this paper tackles an important problem, and shows promising results, though the proposed approach has limited technical novelty. The paper could be further improved with a more comprehensive limitations section and more discussion on the use of likelihood to perform OOD. On this basis I recommend a weak accept.

---

### Official Review · Reviewer_hs3b · 2023-07-18

**Confidence:** 4
**Originality:** Fair
**Technical Quality:** Good
**Clarity Of Presentation:** Good
**Impact:** 3

**Recommendation:**

Weak Accept: I recommend accepting the paper, but will not argue for my recommendation if the majority of other reviewers have a different opinion.

**Review:**

Strengths: The paper is generally clear, technically sound, and is extensively compared to baselines. Moreover, the problem of OOD detection is highly relevant for robot learning today. While the paper essentially combines existing techniques (LARS and IB) for application to class-conditioned flows, the paper could still be useful for future normalizing flow practitioners.

Weaknesses and concerns:
- As a general comment, I am curious how often multimodal distributions with pathologically disconnected supports actually occur in practice for image classification. I ask because the overall OOD accuracy improvement in the real image datasets in Table 2 seems fairly minor. For the purpose of understanding, it would strengthen the paper to analyze the learned acceptance function in these settings: does it actually learn a multimodal base distribution, like in the synthetic examples? How many disconnected modes are there? If the learned base distribution is still unimodal, it would seem that topology is not the bottleneck, but rather that using a modified based distribution may overall improve the training optimization landscape compared to using a standard Gaussian, and maybe the contribution should be stated as such.
- It is well-known that flows can often assign higher likelihood to OOD data than in-distribution data, and thus may have limited utility as an OOD measure [1]; however, this limitation is not discussed in the paper. Intuitively, this is because while maximum likelihood training encourages high likelihood values for training data, nothing constrains the training to assign low values to unseen data. Thus, flows still may be insufficient for OOD detection requirements in the safety-critical applications mentioned in the paper (one alternative is to use distance to training data and calibrated bounds on perception error, e.g., [2]).
- I would recommend putting more details on LARS in the paper (perhaps moving some details from the appendix). How is the acceptance function parameterized? How is T selected?
- In the paper, $p_{\phi,\psi}$ is used in the figure subtitles but the notation is never defined?
- Instead of LARS, what about simply using a mixture of bounded uniform distributions as a base distribution for each class? This would be potentially simpler than LARS, and could capture multimodality without inefficient sampling due to the rejection in LARS?
- It wasn’t clear to me why a class-conditional setup is required for the OOD detection problem, since in Sec 2.3, the OOD score is just computed by marginalizing over all classes anyway. Why couldn’t one just train a flow with a single class for all training images and use that to compute an OOD score for test-time images? Some discussion in the paper would help.
- In the information bottleneck loss, how is epsilon chosen, and how sensitive is the performance to the choice of epsilon?
- It would strengthen the paper to report the accuracy of the other baselines on the real robot experiment.
- For the real robot experiment, the OOD detection performance is overall much worse compared to the Pascal-VOC and MS-COCO datasets (in Figure 5c, comparing TPR for the FPRs seen in Table 2). It would be good to have discussion in the paper as to why this is; is it due to limited dataset size?
- What is considered to be an OOD image and ID image for the real robot experiment - this should be described in the results?

[1] Zhang, Goldstein, and Ranganath. Understanding Failures in Out-of-Distribution Detection with Deep Generative Models. ICML’21.

[2] Chou, Ozay, and Berenson. Safe Output Feedback Motion Planning via Learned Perception Modules and Contraction Theory. WAFR’22.

**Quality Of The Limitations Section:**

Additional details required

**Questions For Rebuttal:**

Please respond to the questions in the review above.

**Robotics Focus:**

Sufficient demonstration on hardware

**Summary Of Paper:**

This paper presents a method for improving the distribution modeling accuracy of normalizing flows when the target distribution has complex topological properties, for the purpose of OOD detection in object classification from images. In particular, a fundamental limitation of normalizing flows is that the learned distribution shares the topological properties of the base distribution; hence, if the base distribution is unimodal (e.g., a Gaussian, which is commonly used), the flow may be unable to accurately model the target distribution. To overcome this, this paper proposes using a more expressive learned base distribution combined with an information bottleneck loss. At test time, the OOD metric used is the predicted likelihood, marginalized over classes. The approach leads to improvements in OOD detection accuracy on pre-existing datasets as well as on images collected on a real robot.

**Summary Of Recommendation:**

Paper is clear with extensive baseline comparisons, but could use more motivation on design choices in the method, discussion of limitations, and clarifications on the real robot experiment.

---

> ### Author Response · Authors · 2023-08-14
> **Response to Reviewer hs3b**
>
> Dear Reviewer hs3b,
>
> we would like to thank the reviewer for the time and effort spent on reviewing our paper. We have carefully examined each comment from you and made corresponding revisions to the orginal draft (in attachment). These suggestions have subtantially improved the quality of the manuscript. Within this reply, we will provide detailed response to your questions and comments in order to fully address your concerns. If you find any aspects that require further clarification, we are eager to address them promptly.
>
> **Q: " why a class-conditional setup is required for the OOD detection problem;  Why couldn’t one just train a flow with a single class for all training images and use that to compute an OOD score for test-time images;".**
>
> **A:** To just train a flow with a single class for all training images is among our baselines in Table 2., e.g., Flows Gaussian, Flows MoG, and Flows RSB. They are unconditional and trained for all training images. The experimental results show that these approaches underperform without considering the class information.
> On the other hand, note that we use a flow for all training images and the class-conditional setup is only for the base distribution,  with which we attempt to improve the OOD performance by incorporating task-specific information such as class labels as condition with the Information Bottleneck loss. To make it more clear, we have updated Figure 1 in the revised manuscript.
>
> **Q: ”How are hyperparameters optimised? Is the approach sensitive to the value of beta in the IB loss? How was this chosen?”.**
>
> **A:** We provide the ablation study on $\beta$ and $\sigma$ of the IB loss in section 6.1 and 6.3 of the supplementary materials. $\beta$ controls the contribution of the class-conditional term, which includes the task-specific semantic class information. From the results, we observe that with too small $\beta$, the class-conditional density is not modelled sufficiently. By increasing  $\beta$ , the performance boosts and gradually plateaus, implying the success of incorporating class-conditional information for more effective OOD detection. On the other hand, along the dimension of $\sigma$, adding a certain level of noise (e.g. 0.5) can improve the performance. However, too much noise (e.g 5.0) corrupts the input and degrades the performance significantly due to excessive information loss. The final values of them are selected based on the performance on a validation set.
>
> **Q: "It would strengthen the paper to report the accuracy of the other baselines on the real robot experiment.".**
>
> **A:** We note that the goal of the real robot deployment is not to compete with other baselines, but to demonstrate the practicality of our method. For the purpose of fair and rigorous comparison against other baselines, we provide the results on public benchmark datasets in Table 2 by strictly following a standardised experimental protocol. As to conduct the real-robot experiments requires more effort, which is hard to finish during the period of rebuttal, we can add them later if required.
>
> **Q: "For the real robot experiment, the OOD detection performance is overall much worse compared to the Pascal-VOC and MS-COCO datasets. It would be good to have discussion in the paper as to why this is; is it due to limited dataset size?".**
>
> **A:** We do not have clear idea why the performance drops after switching the network architecture of the object detector (Faster-RCNN to Yolov7) and the datasets (different ID and OOD distributions). There can be different factors influencing the resultant performance. We hypothesize that the OOD distribution is closer to the ID distribution than the case in the benchmark datasets. This is because we rendered the synthetic images in a highly photorealistic manner. Therefore the features of the ID objects extracted from the object detector should be more similar to the OOD objects, hence leading to the performance drop on OOD detection. We have revised the discussion part correspondingly.
>
> **Q: "What is considered to be an OOD image and ID image for the real robot experiment - this should be described in the results?"**
>
> **A:** We have added more details regarding this experiment, Line 268-272 in the revised manuscript. To briefly explain, we train the detector based on only synthetic images and deploy it in an environment with only real objects. The task is to identify the falsely detected real objects as OOD since they are from the data distribution different to the synthetic ones. We provide some exemplar images of both synthetic and real images in Figure 1. In the supplementary materials for better understanding.
>
> Best regards,
>
> Paper241 Authors

---

### Official Review · Reviewer_q52p · 2023-07-20

**Confidence:** 4
**Originality:** Good
**Technical Quality:** Good
**Clarity Of Presentation:** Good
**Impact:** 2

**Recommendation:**

Weak Accept: I recommend accepting the paper, but will not argue for my recommendation if the majority of other reviewers have a different opinion.

**Review:**

The paper proposes a novel method for ODD detection. The paper does not fully demonstrate its use in robotics. It is unclear whether the problem being addressed indeed exists in robotic inspection problems, or not. It would have helped if this was made clear right at the beginning of the paper. The paper's readability would have improved if a section on the background covered more details about the prior methods used.


**Quality Of The Limitations Section:**

Additional details required

**Questions For Rebuttal:**

The method seems nice, but its application to robotics is not fully demonstrated. Any comments by authors about this would help. Thanks!

**Robotics Focus:**

Relevant but unlikely to deploy to hardware in near future

**Summary Of Paper:**

The paper proposes a new out-of-distribution method using normalizing flows but with a class-conditional base distribution. This enables the new method to ‘topologically match’ the target distribution, thereby enabling better ODD detection.

**Summary Of Recommendation:**

The work is good and sound. However, its use in robotics is not very clear. The robotic inspection and maintenance scenario (Sec 4.3) is not clearly explained. It would have helped the paper if either this was the key application explored, or at least two other robotic applications of the proposed ODD method were shown.

---

### Official Review · Reviewer_X7BF · 2023-07-22

**Confidence:** 3
**Originality:** Fair
**Technical Quality:** Good
**Clarity Of Presentation:** Very Good
**Impact:** 3

**Recommendation:**

Weak Accept: I recommend accepting the paper, but will not argue for my recommendation if the majority of other reviewers have a different opinion.

**Review:**

Strengths:
- The proposed approach balances computational complexity and performance, yielding good density modeling yet remaining practical to deploy in real-time contexts.
- The paper is well written, and the authors provide a good amount of background on the prior advances the authors build on in this work.

Weaknesses:
- The work strikes me as a straightforward application of prior advances for improving the quality of NF density modeling to existing techniques using NFs for OOD detection. With this in mind, the main contribution of the authors is to show that such a technique works well and is efficient enough to run in real-time. Specifically, the authors claim that spending computational budget on resampling base distributions per class is better than just adding more layers to the normalizing flow. This is an interesting claim, and I would have liked to see an empirical evaluation of this tradeoff.
- The real-robot experiment was not described in sufficient detail: how were the images labeled? how were images classified as being in- or out-of-distribution? Furthermore, the real-world experiment lacked comparison to any baselines other than the base YOLO model.

Minor comments
- Some claims are contradictory:
  - Line 106-108: "This way we get similar benefits...without having to burden the cost on marginalization over classes"
  - Line 163: "We marginalize the density over classes"

**Quality Of The Limitations Section:**

Limitations are addressed clearly

**Questions For Rebuttal:**

How are hyperparamters optimized? Is the approach sensitive to the value of beta in the IB loss? How was this chosen?

Could you compare to other non-flow based approaches to OOD detection? Specifically, comparing to the Mahalanobis distance (Lee et al. 2018) and Relative Mahalanobis Distance (Ren et al., 2021) approaches would be worthwhile, since they also aim to detect anomalies at the logit level, and could serve as a point on the compute / accuracy tradeoff curve (as they can be thought of as a limit taking the number of flow layers to 0).

**Robotics Focus:**

Sufficient demonstration on hardware

**Summary Of Paper:**

This work addresses OOD detection based on density modeling with normalizing flows (NFs). Specifically, the authors propose an approach which follows past work in modeling the distribution of logits with a NF. The authors improve on past work aiming to address topology mismatch issues between the logit distribution and base distribution. To do so, they use class conditional distributions, as opposed to a single marginal, and further reshape each class's base distribution using learned accept/reject sampling. They demonstrate their approach on object detection tasks in both simulated and real-robot OOD detection tasks.

**Summary Of Recommendation:**

Overall, this paper tackles an important problem and presents a solution that performs well on the domains tested, but the approach is a relatively straightforward application of existing work, and the experimental evaluation is missing useful comparisons and lacking details regarding the real-robot experiments.

---

> ### Author Response · Authors · 2023-08-14
> **Response to Reviewer X7BF**
>
> Dear Reviewer X7BF,
>
> we would like to thank the reviewer for the time and effort spent on reviewing our paper. We have carefully examined each comment from you and made subtantial revisions to the orginal draft (in attachment). These suggestions have subtantially improved the quality of the manuscript. Within this reply, we will give detailed response to your questions and comments in oder to fully address your concerns.
> We would deeply appreciate if the reviewer could consider rasing the score to a "weak accept" instead of a "weak reject".
> If you find any aspects that require further clarification, we are eager to address them promptly.
>
> **Q: "The real-robot experiment was not described in sufficient detail: how were the images labeled? how were images classified as being in- or out-of-distribution? ".**
>
> **A:** We have added more details regarding this experiment. Please refer to section 4 of the revised manuscript. To briefly explain, we train the detector based on only synthetic images and deploy in an environment with only real objects. The task is to identify the falsely detected real objects as OOD since they are from a data distribution different to the synthetic ones, namely OOD. We provide some exemplar images of both synthetic and real images in Figure 1. of the supplementary materials for better understanding.
>
> **Q: “Furthermore, the real-world experiment lacked comparison to any baselines other than the base YOLO model.”.**
>
> **A:** We note that the goal of the real robot deployment is not to compete with other baselines, but to demonstrate the practicality of our method. For the purpose of fair and rigorous comparison against other baselines, we have provided the results on public benchmark datasets in Table 2 by strictly following a standardised experimental protocol. As to conduct the real-robot experiments requires more effort on time and resources, which is hard to finish during the period of rebuttal, we can add them later if required.
>
> **Q: "Some claims are contradictory: "This way we get similar benefits...without having to burden the cost on marginalisation over classes"; "We marginalise the density over classes".**
>
> **A:** We apologize for this misleading texts. Our claim is actually not contradictory but not ideally phrased. We rephrase this in the revised manuscript. The benefit we mean in Line 106-108 is that we can marginalize over classes in the latent space due to the usage of a conditional base distribution. This only requires one forward pass. In contrast, if we use the mixture of flows and make each member in the mixture class conditional, to marginalise over classes, we need to evaluate all members in the mixture. This requires multiple forwards passes, similar to deep ensemble.
>
> **Q: "Could you compare to other non-flow based approaches to OOD detection? Specifically, comparing to the Mahalanobis distance (Lee et al. 2018) and Relative Mahalanobis Distance (Ren et al., 2021) approaches would be worthwhile, since they also aim to detect anomalies at the logit level".**
>
> **A:** We have added these two baselines in both OOD detection experiments, i.e., the benchmark datasets and the real robot experiment, see Table.2 and  Figure.6.c. in the revised manuscript. Although these two approaches are simple and lightweight, we observe that the performance of Mahalanobis distance (MD) is just on par with the softmax baseline while the relative Mahalanobis distance (RMD) significantly underperforms. This result is also consistent with the findings of the previous work (Ren et al. 2021) for fine-tuned models. They hypothesised that the resulting features are capable of modelling the foreground and background implicitly (without the explicit normalization using RMD). Therefore the effectiveness of RMD in such cases might be limited.
>
> **Q: ”How are hyperparameters optimised? Is the approach sensitive to the value of beta in the IB loss? How was this chosen?”.**
>
> **A:** We provide the ablation study on $\beta$ and $\sigma$ of the IB loss in section 6.1 and 6.2 of the supplementary materials. $\beta$ controls the contribution of the class-conditional term, which includes the task-specific semantic class information. From the results,  we observe that with too small $\beta$, the class-conditional density is not modelled sufficiently. By increasing $\beta$, the performance boosts and gradually plateaus, implying the success of incorporating class-conditional information for more effective OOD detection. On the other hand, along the dimension of $\sigma$, adding a certain level of noise (e.g. 0.5) can improve the performance. However, too much noise (e.g 5.0) corrupts the input and degrades the performance significantly due to excessive information loss. The final values of them are selected based on the performance on a validation set.
>
> Best regards,
>
> Paper241 Authors

---

### Decision · Program_Chairs · 2023-08-30

**Decision:**

Accept (Poster)

**Comment:**

The paper proposes topology-matching normalizing flows for out-of-distribution detection in robot learning. The reviewers had various concerns about the approach including the novelty however it has many positives. The reviewers appreciate the authors' time and effort in addressing the rebuttal, the paper has improved. Therefore, it is my pleasure to let you know the paper is Accepted (Poster). Authors are encouraged to address the reviewer comments to further improve the paper before the publication.